# Agentic Plan Caching: Test-Time Memory for Fast and Cost-Efficient LLM Agents

**Qizheng Zhang**    **Michael Wornow**    **Kunle Olukotun**

Stanford University

## Abstract

LLM-based agent applications have shown increasingly remarkable capabilities in complex workflows but incur substantial costs and latency due to extensive planning and reasoning requirements. Existing LLM caching techniques (like context caching and semantic caching), primarily designed for serving chatbots, are insufficient for agent applications where outputs depend on external data and environmental contexts. We propose **Agentic Plan Caching (APC)**, a novel **test-time memory** that extracts, stores, adapts, and reuses structured plan templates from planning stages of agent applications across semantically similar tasks to reduce the cost and latency of serving. Unlike traditional semantic caching, our system extracts plan templates from completed agent executions at test-time, employs keyword extraction to match new requests against cached plans, and utilizes lightweight models to adapt these templates to task-specific plans with contexts. Evaluation across multiple real-world agent applications shows that our system can reduce costs by 50.31% and latency by 27.28% on average while maintaining performance, offering a more efficient solution for serving LLM-based agents that complements existing LLM serving infrastructures.

## 1   Introduction

Agent applications based on Large Language Models (LLMs) have shown early promise in replicating human performance on a broad range of workflows, from coding [26, 28, 61] to web navigation [23, 70] to open-ended research [3, 8] to social interactions [42, 57]. Many of these LLM-based agents follow a **two-stage pipeline**, often referred to as the *ReAct*-agent loop [63, 45], that alternates between: **(1) Plan** – reasoning about what to do next, and **(2) Act** – executing those plans. While effective, these agents incur significant costs due to the complexity of executed workflows [67, 31] and need to interact with external tools and environments [43]. Specifically, the Plan stage is often implemented via test-time compute techniques [13, 47] like chain-of-thought reasoning [53], which can require numerous LLM queries and access to expensive LLMs (*e.g.,* reasoning models). This results in substantial costs for executing agentic workflows via APIs [15, 39] or locally [35].

To reduce LLM costs, methods have been developed to optimize responses to individual queries [32, 69]. In particular, *caching* has emerged as a popular approach, with two primary implementations: **Context caching** (*e.g.,* KV cache reuse and prompt caching [20, 60, 62]) stores internal model states to speed up subsequent generations, while **semantic caching** [10, 46, 2] stores and reuses (input, output) pairs to accelerate the serving of queries that are similar to historical queries.

These caching techniques, however, have significant limitations when applied to Plan-Act agents. These agents often require making *data-dependent decisions*, *i.e.,* LLM outputs depend on external data or contextual information that varies between runs. For example, in data analysis applications, the same high-level query (*"summarize key statistics of this dataset"*) will result in similar high-level plans, but different specific details depending on the characteristics of the dataset provided. Similarly,

39th Conference on Neural Information Processing Systems (NeurIPS 2025).

in web or GUI navigation tasks, the same high-level query (*"delete the top comment"*) will require similar sequences of actions (*e.g., "click the menu button, scroll down"*), but the specifics may differ depending on screen size and window position (*e.g., "click coordinates (130, 493), scroll down 38 pixels"*). In such cases, conventional caching fails because it does not separate the core intent of the query from the dynamic context. Agents may benefit from local (*i.e.,* individual query-level) optimizations, but miss opportunities for global improvements that leverage patterns across the entire task execution.

To overcome these limitations, we propose **Agentic Plan Caching (APC)**, a novel **test-time memory** that reduces the serving costs of LLM-based agents that follow the Plan-Act paradigm by adapting and reusing prior execution plans across semantically similar workflows. Our key insight is that the Plan stage, which incurs the majority of LLM compute cost, is often repeated (within or across workflows) despite yielding outputs that could be reused in future requests while maintaining performance. When an agent completes an execution of a workflow, we extract structured **plan templates** from the agent execution log. When a similar request arrives, we employ **keyword extraction** to identify the most important semantic target of the query, then match it against the cache to retrieve the most relevant plan template. Our approach differs from semantic caching by avoiding query-based cache lookups, which we found sub-optimal for agent applications. Whenever additional planning is required, we utilize a lightweight model to adapt the cached structured plan template into more detailed plans with task-specific contexts (*e.g.,* fiscal year and company name in financial data-intensive reasoning [39]), rather than employing an expensive model.

Although several memory architectures have been proposed to help agents store and learn from past experiences [49, 40, 51, 59, 65], these efforts primarily focus on using such memories to improve the agent's accuracy on completing workflows (*e.g.,* with fewer hallucinations [9] or with higher task success rate [51]) rather than to reduce the cost of serving the agent. To our knowledge, the use of historical experiences to more efficiently serve LLM-based agents remains underexplored, particularly for applications where outputs depend on input data or environmental conditions external to the query itself.

We evaluate agentic plan caching on five diverse agent workloads and find that it **reduces costs by 50.31% and latency by 27.28% (on average) while maintaining 96.61% of optimal accuracy**. The agentic plan caching we propose is compatible with existing LLM serving and agent frameworks, and can be used jointly with existing caching techniques as well.

In summary, we make the following contributions:

1. **Analysis of Caching Techniques for Serving LLMs:** We conduct a comprehensive analysis of existing caching techniques for LLM serving (context caching and semantic caching), and point out why they are insufficient for the era of agentic AI applications.
2. **Proposal of Agentic Plan Caching:** We propose the idea of agentic plan caching, which shifts the focus from query-level caching (suitable for chatbots) to task-level caching (targeting LLM-based agents). We design and implement a novel caching system that extracts, stores, adapts and reuses agent-generated plans at test-time.
3. **Evaluation of Caching Techniques:** We evaluate our agentic plan caching system on top of real-world agent architecture and five datasets/benchmarks, and find that our approach can reduce cost by 50.31% and latency by 27.28% (on average), while maintaining 96.61% of optimal application performance.

## 2 Background and Motivation

### 2.1 Plan-Act Agents

The rise of large language models (LLMs) has driven the rapid expansion of agentic AI applications. Unlike single-model tasks like chatbots [17], math [24], or coding [16], these applications coordinate multiple models and queries to solve complex tasks, like data-intensive reasoning [39], software engineering [64, 52], web navigation [70], etc.

Many such agentic AI applications, like multi-agent systems [50, 22] and cloud-edge LLM systems [66, 39], follow a two-stage pipeline loop (similar to the ReAct-agent loop [63]), as shown in Figure 1a: (1) Plan and (2) Act. In the Plan stage, a planner LLM generates a strategy (*e.g.,* task

decomposition, information retrieval) that guides subsequent actions of acting LLMs. In the Act stage, the actor LLM acts accordingly based on devised plans and external context or environment, and passes down the response to the planner LLM for the next step.

However, due to the use of multiple LLMs and queries, especially with advanced models like reasoning or multimodal LLMs, these agentic applications can incur significant costs [30, 41], particularly in terms of token ingestion/generation. Optimizing these costs is crucial for scaling agentic AI applications.

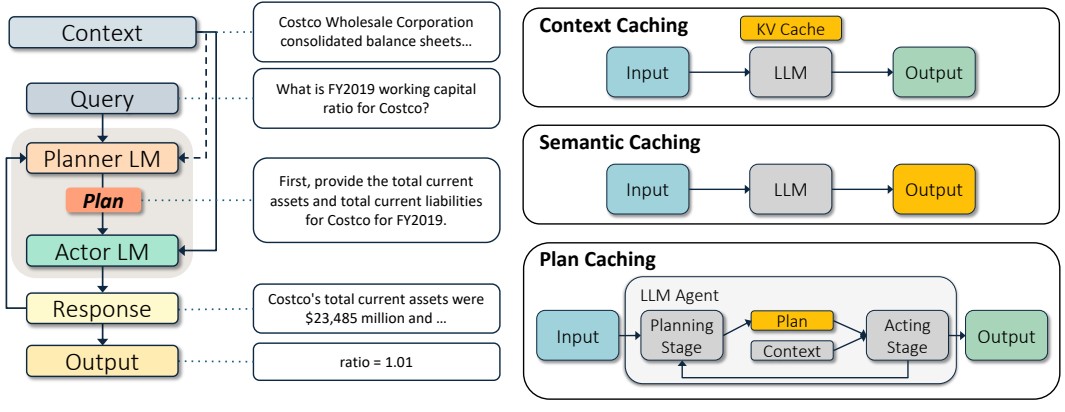

(a) Plan-Act LLM application  (b) Comparison of different LLM caching techniques

Figure 1: **Plan-Act LLM Applications and Caching Techniques.** (a) A typical Plan-Act agent pipeline loop and (b) a comparison of LLM caching methods, with cached components highlighted in yellow.

## 2.2 LLM Caching: Methods and Limitations

**Caching** is one of the most widely-adopted techniques for reducing the serving cost of LLM applications. The goal of caching is to eliminate redundant computation. Context caching [20, 62, 60, 58], also known as KV cache reuse or prompt caching, involves storing and reusing the key-value pairs generated during the prefill phase of LLM inference. Semantic caching [10, 46, 2], on the other hand, stores input-output pairs of previous LLM invocations. This relies on the fact that many prompts share similar underlying intents and thus expected outputs despite having different wording [46].

We find that existing caching techniques, primarily designed for serving **chatbots** (at query-level) instead of **agents** (at task-level), have three major limitations as described below.

**1) Model-Specific Constraints.** Context caching relies on KV cache as the medium for storing and reusing knowledge [34, 20, 60]. These KV caches are inherently **model-dependent** and not easily transferable across different models [56, 33], since even identical text prompts produce model-specific KV caches. While this limitation is negligible for chatbots that consistently use a single model with the same system prompt, it becomes a problem for agentic AI applications that typically employ multiple LLMs across various processing stages.

**2) Data-Dependent Outputs.** Semantic caching stores input-output pairs from previous LLM calls, assuming outputs depend solely on input prompts [10, 46]. While this holds for chatbots, many agentic AI applications are **data-dependent**: Outputs depend not only on input queries but also on external data (*e.g.,* data-intensive reasoning [39]) or dynamic environments (*e.g.,* web or GUI agents [70, 55, 54]). This dependency complicates the reuse of cached responses even when input prompts are semantically similar.

**3) Limited Adaptability.** Both context and semantic caching lack flexibility for handling slight variations in input. Context caching requires exact text matches. Semantic caching, while more accommodating, does not capture the transformation process from prompt to response. This could hinder adaptation to similar queries with minor differences (*e.g.,* numeric values or variable names in mathematical reasoning [18], coding tasks [26]), a common challenge in agentic AI.

### 2.3 Related Work

**Agent Memory**    Prior work has explored augmenting LLM agents with external memory to (1) reduce hallucinations through context-aware responses [9] and (2) enable complex, long-horizon tasks [51]. Some studies focus on defining memory formats [63] and managing memory efficiently [40, 59]. While our caching system can be adapted as a form of agent memory, it diverges by targeting serving cost reduction rather than enhanced capability, a largely unexplored area.

**LLM Serving Engines**    Existing LLM serving engines like vLLM [32] and SGLang [69] optimize general query inference at scale through techniques such as KV cache management and request scheduling. Our approach is compatible with these systems, extending their capabilities to incorporate cost-effective caching for agentic AI scenarios.

**Case-Based Planning**    Case-Based Planning (CBP) [11, 48, 12] is a problem-solving paradigm where new plans are produced by adapting previously solved cases rather than constructing plans from scratch. By exploiting similarities between past and current situations, CBP supports efficient plan reuse, continual learning, and incremental refinement. While our work shares the high-level intuition of "plan storage and reuse", it targets a fundamentally different setting from classic symbolic CBP: LLM-based, neural agents performing open-ended natural-language actions. Specifically, our system (1) automatically extracts reusable plan templates at test time from unconstrained LLM generations, instead of relying on hand-crafted symbolic plans, and (2) leverages these stored plans for cost-efficient inference in LLM agents.

## 3    The Agentic Plan Caching (APC) Framework

In this section, we provide an end-to-end overview of our agentic plan caching framework (§3.1) and then discuss the motivations behind key design choices (§3.2).

### 3.1    Overview of System Design

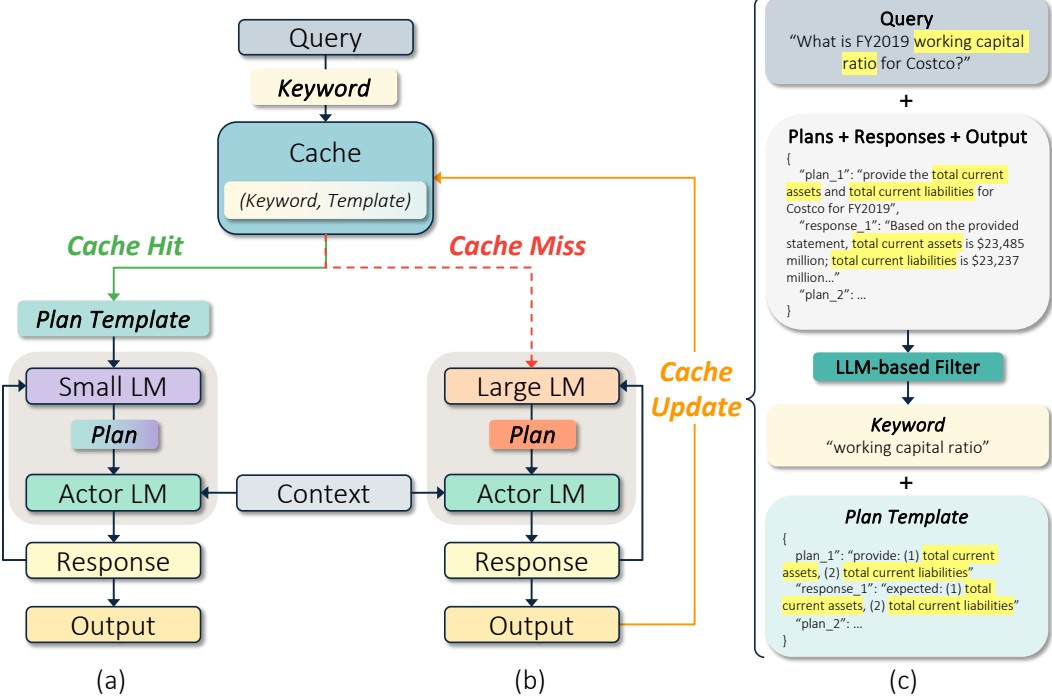

Figure 2: **Agentic Plan Caching Framework.** We show: (a) cache hit workflow, (b) cache miss workflow, and (c) plan template generation for new cache entries.

We provide an end-to-end overview of the agentic plan caching framework in Figure 2. The process begins with a cost-effective language model (*e.g.,* GPT-4o-mini) extracting a keyword that captures the higher-level intent of the input task query (*e.g., "compute the average of all numbers listed in an external document" →"mean calculation"*). This keyword is then used to search the plan cache, which stores (keyword, plan template) pairs, potentially resulting in a cache hit or miss.

For a cache hit – Figure 2(a) – , a small planner LM ("Small LM") adapts the retrieved plan template for the current execution by incorporating context-specific information (*e.g.,* user information, environment variables). For a cache miss – Figure 2(b) – , a large planner LM ("Large LM") generates a new plan from the input task query. The adapted or generated plan, along with the task context (*e.g.,* external data or web/GUI environment), is then passed to the "Actor LM", which produces a response. The response is evaluated by the "Planner LM" to determine if further iterations are needed. If the task is complete, the final output is generated, concluding the agent's execution.

In the case of a cache miss, once the agent successfully completes execution with correct outputs, the system generates a plan template that can be reused in future invocations of the agent through the following two-step process: (1) A rule-based filter extracts critical information from the execution log while discarding irrelevant details, such as verbose reasoning steps; (2) A lightweight LLM-based filter removes context-specific elements (*e.g.,* entity names, numeric values), producing a generalized template ("Plan Template") and relevant keywords for caching (Figure 2(c)).

Additional algorithmic details are provided in the Appendix.

## 3.2 Design Choices

**Why Keyword Extraction?**   A common method for identifying similar queries in a cache is to assess semantic or textual similarity, as seen in frameworks like GPTCache [10] which use embeddings for similarity searches. However, we find that *query-based similarity matching, despite its popularity, is insufficient for detecting cache hits/misses for agentic plan caching*. This is because it might overemphasize context-specific details (*e.g.,* names of individuals or companies) rather than the broader intent of queries, which makes it difficult to establish an effective similarity threshold [46]. This often results in a high number of false positives (irrelevant cache hits) or false negatives (missed reuse opportunities). In contrast, extracting keywords that reflect the higher-level intent of queries provides a more reliable indicator of whether two queries would result in similar agentic plans, as illustrated in Figure 3.

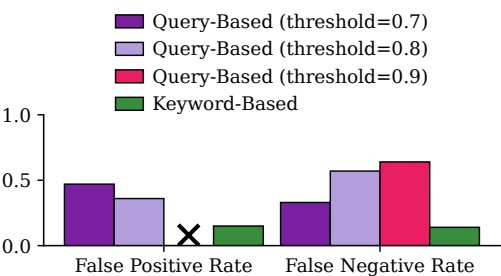

Figure 3: **Query-Based v. Keyword-Based Cache Search.** Keyword-based cache search achieves lower levels of false positive and false negative rates than query-based similarity cache search across different thresholds. This suggests that semantic similarity of queries alone may not effectively capture shared task intents and reusable plans.

**Exact Matching v. Fuzzy Matching**   Our system uses exact matches (between keywords) to minimize false positives. While fuzzy search [27] (identifying cache hits based on similar but not identical keywords) could handle approximate key similarities and is feasible to integrate, we opted against it and leave it for future exploration for two main reasons: (1) Determining fuzzy matches based on semantic or textual similarity of keywords would reintroduce challenges faced by semantic caching, such as setting effective similarity thresholds, and (2) although lightweight LMs could potentially enable fuzzy matching, cache lookups must remain fast and cost-effective, particularly

in low-hit-rate scenarios. We present an in-depth analysis of the overhead and scalability of fuzzy matching in §4.4.

**Caching Plan Template v. Caching Full Execution History**   One naive approach to reuse historical experience is to cache and reuse past agent execution logs (containing all inputs and outputs from planner and actor LMs) as in-context learning examples for the small planner LM. However, in our experiments (§4.2), we find that small planner LMs, usually based on small language models (*e.g.,* we use LLaMa-3.1-8B), struggle to handle long-context and unfiltered agent execution logs even when containing reusable plan information. This motivates us to filter agent execution logs into high-quality plan templates, and re-adapt them so that small planner LMs can better take advantage of their information.

## 4   Results and Evaluation

We evaluate our agentic plan caching (APC) framework on five agent workloads that span a diverse range of data-intensive reasoning and agentic capabilities. These include long-context data reasoning (FinanceBench [25], QASPER [19]), mathematical reasoning (Tabular Math Word Problems [36], AIME 2024 and 2025 [37]), and multi-step agentic reasoning and tool use (GAIA [38]).

Our key findings are:

- **Reduced Cost:** APC reduces agent serving costs by an average of 50.31% (§4.2).
- **Reduced Latency:** APC reduces agent serving latency by an average of 27.28% (§4.3).
- **High Accuracy:** APC maintains 96.61% of application-level performance compared to the accuracy-optimal baseline (§4.2).
- **Low Overhead:** On average, keyword extraction and cache generation account for only 1.04% of the overall cost of running each benchmark (§4.3).

### 4.1   Experiment Setup

Our agentic plan caching system is built on the Minion architecture (Figure 1a) from the Minions project [39], a sequential Plan-Act LLM framework that can be readily generalized. The Minion architecture is composed of a large (cloud-hosted) planner LM for reasoning and task decomposition and a smaller (locally hosted) actor LM with access to additional context for plan execution. Given a task query, the planner LM and the actor LM collaborate iteratively (as in Figure 1a) to produce a final output. We set the maximum number of iterations to be 10.

While we adopt this architecture for clarity, APC is not restricted to Minion-like agent architecture; we demonstrate its integration into other agent architectures and report end-to-end results in §4.2. Additional implementation details and dataset specifications are provided in the Appendix.

**Evaluation Metrics**   We assess application-level performance using GPT-4o as the evaluation model, as LLM-based evaluation is more effective than exact matches or F1 scores for numeric evaluation and long-form responses [14, 21, 68]. Cost is calculated based on input/output tokens and the latest API pricing from commercial LLM providers (OpenAI API [7] and TogetherAI API [1]). Additional details on evaluation models, prompts, and API pricing are included in the Appendix.

**Language Models**   For the main results (§4.2), we use GPT-4o [5] as the planner LM and LLaMa-3.1-8B [6] as both the small planner LM and actor LM. For keyword extraction and cache generation, we use GPT-4o-mini [4]. To demonstrate broader applicability, we include a sensitivity analysis with a wider range of models in the Appendix.

**Baselines**   We evaluate our system against the following baselines:

- **Accuracy-Optimal**: No caching is applied. The large planner LM is always used to establish the best achievable application performance.
- **Cost-Optimal:** No caching is applied. The small planner LM is consistently used to assess the lowest possible cost.

- **Semantic Caching**: We implement a query-level semantic caching method based on previous work [10, 46]. Following the approach of GPTCache [10][1], we cache and reuse responses to individual queries, determining cache hits based on query-level similarity. We set similarity thresholds to be 80%, 85%, and 90%; a lookup is considered a hit if the query-level similarity is above this threshold.
- **Full-History Caching** (discussed in §3.2): Inspired by knowledge caching in retrieval-augmented generation [62, 29], this baseline caches the complete agent execution log, including inputs and outputs of all LLM agent components. Cache hits are determined by keyword-level similarity. Upon a hit, the cached execution log is used as an in-context example for the small planner LM to generate new plans.

## 4.2 Main Results

As shown in Figure 4 and Table 1, agentic plan caching reduces cost by 50.31% on average while maintaining 96.61% of application-level performance compared to the accuracy-optimal baseline. We note that:

- **Semantic Caching:** Despite cost savings at lower similarity thresholds, semantic caching suffers from a high rate of false-positive cache hits, leading to substantial performance degradation. Additional case studies of false-positive hits are provided in the Appendix.
- **Full-History Caching:** While full-history caching preserves past plans and actions that might help plan generation for similar tasks, it underperforms agentic plan caching in accuracy (72.00% vs. 85.50% in FinanceBench) and incurs higher costs ($1.99 vs. $1.86). This is due to the small planner LM's difficulty in processing lengthy and unfiltered histories, emphasizing the necessity of our LLM-based filter to extract concise, reusable plan templates.

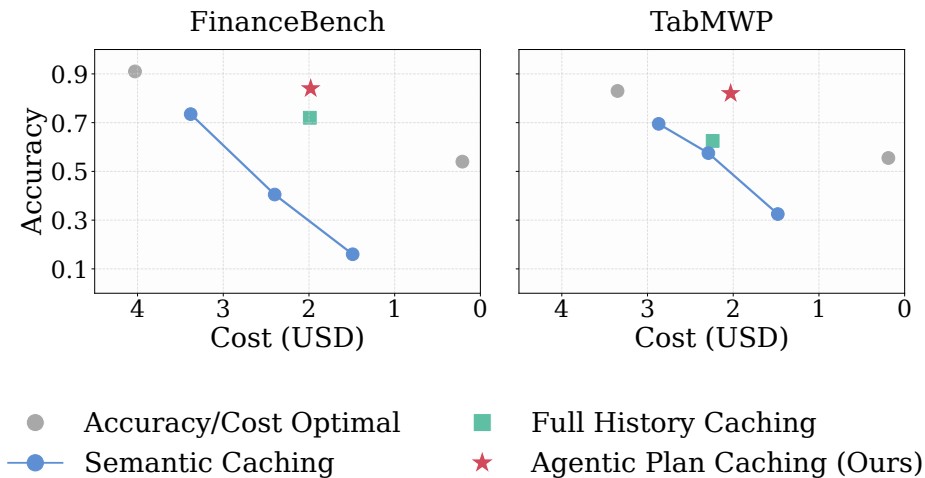

Figure 4: **Results across Four Baselines and Agentic Plan Caching.**

**Results on GAIA with Open Deep Research Agent**   Beyond Minion-based architectures, we integrate APC into the Open Deep Research agent from the Hugging Face `smolagents` library [44] using GPT-4o as the large planner LM and GPT-4o-mini as the small planner LM. As shown in Table 1, on the GAIA benchmark [38], APC achieves a 76.42% reduction in cost (from $69.02 to $16.27) with only a 0.61% drop in accuracy (37.58% to 36.97%), demonstrating strong cost-efficiency even in complex, open-domain agent settings.

A closer analysis reveals that GAIA's heterogeneous task space, ranging from video dialog reasoning to sales computation, limits the effectiveness of keyword-based cache retrieval, as many task descriptions are highly specific and rarely recur. Despite fewer cache hits during initial planning,

---

[1]We do not use the official GPTCache release as it (1) lacks support for post-GPT-4 OpenAI models and (2) relies on a deprecated version of the OpenAI API.

| Method | Minion | | | Open Deep Research |
| --- | --- | --- | --- | --- |
| | QASPER | AIME 2024 | AIME 2025 | GAIA |
| | Cost↓ / Accuracy↑ | Cost↓ / Accuracy↑ | Cost↓ / Accuracy↑ | Cost↓ / Accuracy↑ |
| Accuracy-Optimal | $2.14 / 58.00% | $1.14 / 64.52% | $1.34 / 61.29% | $69.02 / 37.58% |
| Cost-Optimal | $0.21 / 53.00% | $0.65 / 48.39% | $0.60 / 48.39% | $3.16 / 19.39% |
| APC (Ours) | $0.78 / 57.00% | $0.85 / 61.29% | $0.81 / 58.06% | $16.27 / 36.97% |

Table 1: **More Results.** We evaluate APC on a diverse set of benchmarks covering reasoning and agentic capabilities, as well as agent architecture like Minion and Open Deep Research.

APC improves efficiency in re-planning phases by reusing prior plan structures, thereby reducing redundant large-model invocations.

**Cache-Miss v. Cache-Hit Accuracy**   To assess the impact of caching on application performance, we compare cache-miss and cache-hit accuracy across semantic caching, full-history caching, and agentic plan caching (Figure 5). For semantic and full-history caching, cache-hit accuracy is significantly lower than cache-miss accuracy, indicating a performance trade-off despite potential cost savings. In contrast, agentic plan caching maintains consistent accuracy regardless of cache-use status, demonstrating its ability to preserve application performance without degradation.

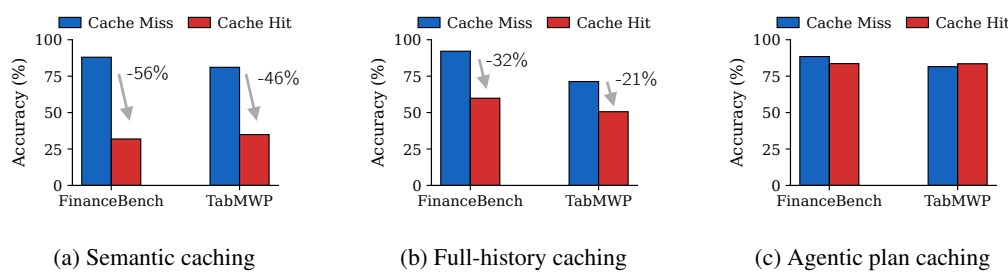

(a) Semantic caching          (b) Full-history caching          (c) Agentic plan caching

Figure 5: **Accuracy Comparison across Caching Methods.** While semantic caching with threshold=0.9 in (a) and full-history caching in (b) experience notable accuracy drops during cache hits, agentic plan caching in (c) maintains stable performance across datasets.

### 4.3   Cost and Speed Analysis

**Cost Breakdown**   We analyze the additional overhead introduced by the agentic plan caching mechanism through a cost breakdown analysis (Table 2). On average, keyword extraction and cache generation account for only 1.04% of the total cost. This minimal overhead is achieved because: (1) extracting higher-level goals or intents from task queries can be effectively handled by lightweight models at the scale of GPT-4o-mini or smaller, and (2) cache generation leverages rule-based methods to extract templates and uses a lightweight language model only for filtering out query-specific or context-specific details, which is a task well-suited to compact models.

**Worst-Case Cache Overhead**   We assess the overhead incurred under the worst-case scenario, where the cache hit rate is zero. As shown in Table 2, even in this scenario, the cost from keyword extraction and cache generation is minimal (1.31% on average). In practical deployment, a potential mitigation strategy is to dynamically disable caching when hit rates remain persistently low.

**Latency Analysis**   We evaluated the wall-clock latency of our system and compared it to both accuracy-optimal and cost-optimal baselines. Additionally, we provide a detailed breakdown of latency incurred by each component in our plan caching pipeline. This microbenchmark is based on 100 randomly sampled queries from the FinanceBench dataset, with a cache hit rate of 46% (*i.e.,* cached plans were used for 46 of the 100 queries). As shown in the Table 3, APC reduces end-to-end latency by 27.28% on this workload. Most of the additional latency in our system comes from LLM-powered cache generation, which takes an average of 3.99 seconds per entry. To further

| Component | FinanceBench | | TabMWP | |
|---|---|---|---|---|
| | Main Results | Worst Case | Main Results | Worst Case |
| Large Planner LM | $1.7544 (94.17%) | $3.9227 (97.36%) | $1.9823 (97.76%) | $3.3292 (98.33%) |
| Small Planner LM | $0.0168 (0.90%) | – | $0.0095 (0.47%) | – |
| Actor LM | $0.0705 (3.78%) | $0.0529 (1.31%) | $0.0170 (0.84%) | $0.0128 (0.38%) |
| **Cache Overhead** | **$0.0213 (1.15%)** | **$0.0535 (1.33%)** | **$0.0190 (0.93%)** | **$0.0438 (1.29%)** |
| - Keyword Extraction | $0.0050 (0.27%) | $0.0050 (0.13%) | $0.0025 (0.12%) | $0.0025 (0.07%) |
| - Cache Generation | $0.0163 (0.88%) | $0.0485 (1.20%) | $0.0165 (0.81%) | $0.0413 (1.22%) |
| Total | $1.8630 (100%) | $4.0291 (100%) | $2.0278 (100%) | $3.3858 (100%) |

Table 2: **Cost Analysis.** We show the breakdown of agentic plan caching costs, including main results and worst-case overhead (*i.e.,* zero cache hit rate).

| Method | Plan | Act | Keyword Extraction | Cache Lookup | Cache Generation | Total |
|---|---|---|---|---|---|---|
| Accuracy-Optimal | 1813.41 | 94.39 | – | – | – | 1959.24 |
| Cost-Optimal | 856.75 | 93.31 | – | – | – | 1004.79 |
| APC (Ours) | 1011.82 | 131.44 | 42.29 | <1 | 215.80 | 1424.82 |

Table 3: **Latency Analysis.** Breakdown of wall-clock latency across components of the agent pipeline. All values are measured in seconds.

mitigate this overhead, we (1) automatically disable caching when the hit rate is consistently low, and (2) are actively exploring optimizations such as parallel cache generation and speculative next-query inference as part of future work. To summarize, our system offers a favorable performance trade-off: We achieve significantly lower cost than the accuracy-optimal baseline while preserving high accuracy at the cost of moderate latency, most of which is attributable to one-time cache generation.

## 4.4 Scalability and Cache Management

**Effect of Cache Size**   Increasing cache size generally reduces both cost and latency, up to a point of diminishing returns. On the FinanceBench dataset, larger caches yield higher hit rates and lower end-to-end latency, as fewer entries need to be regenerated after eviction. Table 4 reports results using a simple LRU eviction policy. Beyond a certain capacity, approximately exceeding the number of unique task keywords, further enlarging the cache offers minimal benefit. In practice, users can tune cache capacity based on the desired trade-off between speed and storage.

| Cache Size | Hit Rate | Cost | Accuracy | Planning Latency (s) | Cache Gen. Latency (s) | Total Latency (s) |
|---|---|---|---|---|---|---|
| 1 | 2% | $3.97 | 92.00% | 1638.85 | 383.87 | 2232.76 |
| 10 | 13% | $3.51 | 88.00% | 1381.89 | 334.61 | 1911.95 |
| 20 | 28% | $2.95 | 85.00% | 1248.06 | 289.36 | 1772.61 |
| 50 | 45% | $1.88 | 86.00% | 1015.12 | 204.99 | 1459.92 |
| 100 | 46% | $1.86 | 85.50% | 1011.81 | 215.80 | 1424.82 |

Table 4: **Effect of Cache Size.** Larger caches improve hit rate and reduce overall cost and latency until reaching a point of diminishing returns.

**Exact Matching v. Fuzzy Matching**   For exact match lookups, our cache uses Python's built-in dictionary, which provides highly optimized $O(1)$ average-case lookup and insertion. To evaluate empirical performance, we measured wall-clock latency for cache hits and misses across varying cache sizes. Each measurement was averaged over 100 trials with CPU caches cleared before every run. As shown in Table 5, exact matching maintains consistently low latency up to $10^6$ entries.

In contrast, fuzzy matching introduces substantial cache lookup latency that scales poorly with cache size. We implement fuzzy keyword matching in our prototype and use a semantic-similarity model (`SentenceTransformer('all-MiniLM-L6-v2')`). As shown in Table 5, this approach is orders of magnitude slower than exact matching, confirming the computational overhead of semantic search.

| Cache Size | Exact Matching | | Fuzzy Matching | |
|---|---|---|---|---|
| | Cache Hit Latency ($\mu$s) | Cache Miss Latency ($\mu$s) | Cache Hit Latency ($\mu$s) | Cache Miss Latency ($\mu$s) |
| $10^2$ | 13 | 14 | 57 | 24 |
| $10^3$ | 15 | 15 | 75 | 70 |
| $10^4$ | 16 | 17 | 581 | 554 |
| $10^5$ | 22 | 18 | 10388 | 10317 |
| $10^6$ | 56 | 37 | 148449 | 148147 |

Table 5: **Cache Lookup Scalability.** Fuzzy matching incurs much higher latency and scales poorly compared to exact matching. Averages over 100 trials; similarity threshold set to be 0.8.

| Similarity Threshold | Hit Rate | Cost | Accuracy | Planning Latency (s) | Total Latency (s) |
|---|---|---|---|---|---|
| $= 100\%$ | 46% | $1.86 | 85.50% | 1011.82 | 1424.82 |
| $> 80\%$ | 54% | $1.15 | 83.00% | 875.31 | 1219.73 |
| $> 60\%$ | 64% | $0.93 | 77.00% | 720.29 | 1044.50 |

Table 6: **Fuzzy Keyword Matching Results.** Lower similarity thresholds increase hit rate and reduce cost and latency, but degrade accuracy.

We also find that under fuzzy keyword matching, lowering the similarity threshold increases the cache hit rate and reduces cost and latency, but at the expense of accuracy (Table 6). This highlights the inherent trade-off in fuzzy matching: While more aggressive matching improves efficiency, it risks introducing less relevant cached plans. Our cache interface remains flexible—users can enable fuzzy matching and tune thresholds according to their application's tolerance for semantic drift and latency-accuracy trade-offs.

## 4.5 Cold Start

Cold start is an inherent limitation of *test-time* plan caching (as opposed to offline caching), since the cache begins empty. In the early phase, APC experiences higher latency and cost due to frequent cache misses and the need to generate new entries. To quantify this effect, we perform a time-series analysis of cache warm-up, shown in Table 7. As the cache grows, hit rate steadily increases, leading to lower marginal cost and latency over time. In practice, if the target workload is known in advance, users can mitigate cold-start overhead by pre-populating the cache with offline samples before deployment.

| Query Percentile | # Cache Entries | Hit Rate | Cost | Planning Latency (s) | Cache Gen. Latency (s) | Total Latency (s) |
|---|---|---|---|---|---|---|
| 20th | 15 | 14.29% | $0.59 (32.07%) | 260.19 (27.43%) | 59.19 (27.64%) | 358.71 (25.72%) |
| 40th | 27 | 24.39% | $0.97 (52.72%) | 484.88 (51.12%) | 130.29 (60.86%) | 689.49 (49.44%) |
| 60th | 36 | 36.07% | $1.20 (65.22%) | 638.12 (67.28%) | 167.89 (78.41%) | 926.82 (66.46%) |
| 80th | 42 | 40.75% | $1.63 (88.59%) | 820.94 (86.56%) | 195.59 (91.35%) | 1183.39 (84.86%) |
| 100th | 46 | 48.00% | $1.84 (100.00%) | 948.36 (100.00%) | 214.11 (100.00%) | 1394.56 (100.00%) |

Table 7: **Cold Start Behavior.** As the cache warms up, hit rate increases and marginal cost and latency decrease. Pre-warming the cache with offline samples can further mitigate cold-start overhead.

## 5 Conclusion

We introduce **Agentic Plan Caching (APC)**, which shifts the focus from query-level caching (suitable for chatbots) to task-level caching (targeting LLM-based agents). Unlike traditional semantic caching, APC extracts plan templates from completed agent executions at test time, uses keyword-based retrieval to match new queries to cached plans, and leverages lightweight models to adapt these templates into context-specific task plans. By implementing agentic plan caching and evaluating it on five diverse agent workloads and two Plan-Act agent architectures, we demonstrate that our approach reduces agent serving costs by 50.31% and latency by 27.28% (on average) while maintaining 96.61% of optimal application performance. Furthermore, the overhead introduced by plan caching remains minimal, accounting for only 1.04% (on average) of the total serving cost.

# 6 Acknowledgment

We thank all the anonymous reviewers and the area chair for their insightful feedback and suggestions, which significantly enhanced the quality of this work. Qizheng Zhang is supported in part by NSF CNS-2211384. Michael Wornow is supported by the NSF Fellowship, a Stanford HAI Graduate Fellowship, and Stanford Healthcare. We thank Gerry Wan for his contribution to the Open Deep Research experiments in this paper after the initial submission. We thank Avanika Narayan for helpful discussions on Minions, and thank Hanchen Li, Zhiqiang Xie, Jon Saad-Falcon, Azalia Mirhoseini, and the LMCache team for helpful discussions on the project idea and its presentation.

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

# A  Extended Discussion of Methods

## A.1  Algorithms for Framework Design

In this section, we provide more algorithmic details of the agentic plan caching framework. To start with, the end-to-end workflow is provided in Algorithm 1.

---

**Algorithm 1** Agentic Plan Caching: End-to-End Framework

---

**Require:** Query $q$, Context $ctx$, Cache $C$
**Ensure:** Output $o$, Updated Cache $C'$
 1: $keyword \leftarrow$ ExtractKeyword($q$)                          ▷ Extract keyword using a small LM
 2: **if** $keyword \in C$ **then**                                  ▷ Cache hit (Figure 2(a))
 3:    $o, C \leftarrow$ HandleCacheHit($q, ctx, C[keyword], C$)        ▷ Algorithm 2
 4: **else**                                                       ▷ Cache miss (Figure 2(b))
 5:    $o, C' \leftarrow$ HandleCacheMiss($q, ctx, keyword, C$)        ▷ Algorithm 3
 6: **end if**
 7: **return** $o, C'$                          ▷ Return response and possibly updated cache

---

The case of cache hit is demonstrated in Algorithm 2.

---

**Algorithm 2** Cache Hit

---

**Require:** Query $q$, Context $ctx$, Plan Template $template$, Plan Cache $C$
**Ensure:** Output $o$, Cache $C$
 1: $responses \leftarrow \emptyset$                          ▷ Initialize actor LM response to be empty
 2: $plan, o \leftarrow$ LightLM($q, template, responses$)          ▷ Adapt the retrieved template to be a
    task-specific plan using a lightweight model
 3: **Assert:** $o$ is None
 4: **while** $o$ is None **do**
 5:    $response \leftarrow$ ActorLM($q, ctx, plan$)               ▷ Execute the plan based on context
 6:    $responses \leftarrow responses \cup response$
 7:    $plan, o \leftarrow$ LightLM($q, template, responses$) ▷ Generate the final output or a new adapted
    plan
 8: **end while**
 9: **return** $o, C$

---

The case of cache miss is demonstrated in Algorithm 3.

---

**Algorithm 3** Cache Miss

---

**Require:** Query $q$, Context $ctx$, Plan Template $template$, Plan Cache $C$
**Ensure:** Output $o$, Updated Cache $C'$
 1: $log \leftarrow \emptyset$                                  ▷ Initialize the execution log to be empty
 2: $responses \leftarrow \emptyset$                          ▷ Initialize actor LM response to be empty
 3: $plan, o \leftarrow$ PlannerLM($q, responses$)                 ▷ Generate initial plan with full model
 4: **Assert:** $o$ is None
 5: **while** $o$ is None **do**
 6:    $response \leftarrow$ ActorLM($q, ctx, plan$)               ▷ Execute the plan based on context
 7:    $responses \leftarrow responses \cup response$
 8:    $log \leftarrow log \cup \{(plan, ctx, response)\}$                        ▷ Update the log
 9:    $plan, o \leftarrow$ PlannerLM($q, responses$)            ▷ Generate the final output or a new plan
10: **end while**
11: $log \leftarrow log \cup \{o\}$                                              ▷ Update the log
12: $template \leftarrow$ GenerateTemplate($log, keyword$)      ▷ Create reusable plan template based on
    execution log
13: $C' \leftarrow C$
14: $C'[keyword] \leftarrow template$                                        ▷ Store template in cache
15: **return** $o, C'$

---

# B  Extended Description of Experiment Setup

## B.1  Platform

The prototype of our agentic plan caching framework, which we use to run our experiments, is implemented on a Runpod server with dual-socket Intel Xeon Gold 6342 CPUs (96 vCPUs, 2.80GHz base clock, 3.5GHz max turbo) and 512MB total L1, 60MB L2, and 72MB L3 cache. The server supports AVX-512 and runs in a 2×48-core NUMA configuration. For memory, the server is equipped with 503GB of system RAM and no swap space.

## B.2  LLM API Usage and Pricing

All language model inferences in our prototype are performed via third-party APIs. While it is feasible to run inference locally when model weights are available, we use API access to quantify cost in dollar terms for this study. If metrics such as latency or throughput were preferred, running all inferences locally would help eliminate variability introduced by external services, especially when they are hosted remotely. We use the Python APIs for OpenAI (v1.74.0), Together AI (v1.5.8), and Anthropic (v0.49.0). For all experiments, we set `temperature` to 0 (if supported) and `max_tokens` to 4096. Table 8 lists the per-token pricing of all models used in our experiments at the time of evaluation.

| Model Name | API Provider | $ / Million Input Tokens | $ / Million Output Tokens |
|---|---|---|---|
| GPT-4o (`gpt-4o`) | OpenAI | 2.50 | 10.00 |
| GPT-4o-mini (`gpt-4o-mini`) | OpenAI | 0.15 | 0.60 |
| Claude 3.5 Sonnet (`claude-3-5-sonnet-20240620`) | Anthropic | 3.00 | 15.00 |
| Llama-3.1-8B (`Meta-Llama-3.1-8B-Instruct-Turbo`) | Together AI | 0.18 | 0.18 |
| Llama-3.2-3B (`Llama-3.2-3B-Instruct-Turbo`) | Together AI | 0.06 | 0.06 |
| Qwen-2.5-7B (`Qwen2.5-7B-Instruct-Turbo`) | Together AI | 0.30 | 0.30 |

Table 8: **LLM API pricing used in our experiments.**

## B.3  Datasets

**FinanceBench.** We use an augmented version of the FinanceBench test split from HuggingFace[2]. Following the Minions project, we filter for numerical reasoning questions and randomly sample 200 questions for evaluation. Each question requires long-context financial reasoning and is paired with a company-specific document essential for answering. The planner LM does not have access to the financial document, while the actor LM does.

**TabMVP.** We sample 200 questions from the test split of the TabMVP dataset provided by the authors[3]. Each question involves numeric reasoning and is paired with a required table; the question cannot be answered without the associated tabular data. The planner LM does not have access to the tabular data, while the actor LM does.

## B.4  Prompts

### B.4.1  Agent Prompts

We use the same prompts from the Minion protocol in the Minions project [39].

### B.4.2  LLM-as-a-Judge Prompt

As discussed in the results section (§4), standard metrics like exact match or F1 score are often inadequate for evaluating numeric or long-form responses. For LLM-as-a-judge evaluation, we provide the prompt used to assess answer correctness. We closely follow the FinanceBench dataset's original evaluation criteria and define rules for acceptable numeric deviations according to what the

---

[2]`https://huggingface.co/datasets/virattt/financebench`
[3]`https://github.com/lupantech/PromptPG/blob/main/data/tabmwp/problems_test1k.json`

FinanceBench dataset paper proposes, specifying what qualifies as a correct answer. These rules are applied consistently across both FinanceBench and TabMWP evaluations.

**Correctness Evaluation Prompt:** You are a judge that grades numeric answers to data-intensive reasoning problems.
This is the question: {task}.
This is the reference answer: {gt_answer}.
This is the answer given by a language model: {response}.
Please grade it. Requirements:
(1) Please allow minor deviations, such as
(i) giving the answer in billions when the unit was given in the question as millions.
(ii) giving the answer in percentage when the ground truth answer is floating point.
Please also allow small rounding errors or small numerical errors.
(2) Incorrect answers vary, from calculations that are off by small margins to several orders of magnitude, and from making up legal information to giving the wrong direction for an effect (e.g. reporting negative growth when it is actually positive).
(3) Just answer '1' for correct answers, or '0' for incorrect answers.

### B.4.3 Keyword Extraction Prompt

**Keyword Extraction Prompt:** Can you help me summarize what is the 'task' or 'keyword' describing the higher-level goal or intent of this query? Please answer only with the task / keyword, which must be independent from problem-specific details.
{query}

### B.4.4 Cache Generation Prompt

**Cache Generation Prompt:** You will see a filtered JSON trace that shows the complete workflow of how a planner language model solves a complex task by collaborating with an actor language model. Clean up the element of each item in the workflow, so that we can reuse this trace as a reference template (independent from problem-specific variables like company name or fiscal year) when we meet similar tasks later.
Requirements:
(1) the first element in each "workflow" item can only be "message", "output", or "answer",
(2) the task and the workflow should not contain problem-specific details or numbers, and
(3) return the result in JSON format that can be parsed by Python's json.loads().
IMPORTANT: The workflow must maintain the sequence of message->loop(output->message/answer) to ensure proper functioning. Always start with a "message" and end with an "answer".
JSON trace: {trace}

### B.4.5 Cache Adaptation Prompt

**Cache Adaptation Prompt:** You are an intelligent language model that works with another model to solve complex tasks, like data-intensive reasoning questions.
Please construct a follow-up action plan (in the form of a message) based on the task and the reference template.
Reference task: {cached_task}
Reference follow-up action plan (as a message): {next_item_in_cached_template}
Your task is to adapt the reference follow-up message to the current context, maintaining the same inquiry structure but customizing it for the specific details of the current question and model output. Make sure the message asks for information not contained in past messages.
Format your response as a JSON object with a "reasoning" field set to "N/A" and a "message" field containing your action plan message.
Current task: {task}
Past action plans (as messages): {past_messages}
Past actor responses: {past_actor_responses}
Current message:

# C  Extended Results

We evaluate the robustness of agentic plan caching under different choices of large planner LMs, small planner LMs, and actor LMs. Our key findings are:

- **Consistent Gains Across Models:** Agentic plan caching consistently reduces cost and maintains high accuracy across a variety of model choices, beyond those presented in §4.
- **Model Selection Matters:** Despite consistent gains of our method, choosing the right model remains crucial. For example, in most cases, Claude 3.5 Sonnet outperforms GPT-4o in accuracy as the large planner LM but incurs significantly higher cost (Table 9). Similarly, smaller or cheaper models do not always yield better accuracy-cost tradeoffs. For example, using Llama-3.2-3B as the actor LM often leads to both higher cost and lower accuracy compared to Llama-3.1-8B due to insufficient response quality that triggers more Plan-Act iterations (Table 11).

| Method | Large Planner LM | Small Planner LM | Actor LM | FinanceBench | TabMWP |
|---|---|---|---|---|---|
| | | | | Cost↓ / Accuracy↑ | Cost↓ / Accuracy↑ |
| Accuracy-Optimal | GPT-4o | - | Llama-3.1-8B | $4.03 / 91.00% | $3.35 / 83.00% |
| Accuracy-Optimal | Claude 3.5 Sonnet | - | Llama-3.1-8B | $5.77 / 94.50% | $5.09 / 85.50% |
| Cost-Optimal | Llama-3.1-8B | - | Llama-3.1-8B | $0.21 / 54.00% | $0.19 / 55.50% |
| Cost-Optimal | Llama-3.2-3B | - | Llama-3.1-8B | $0.09 / 63.00% | $0.08 / 57.00% |
| Full-History Caching | GPT-4o | Llama-3.1-8B | Llama-3.1-8B | $1.99 / 72.00% | $2.24 / 62.50% |
| Full-History Caching | Claude 3.5 Sonnet | Llama-3.1-8B | Llama-3.1-8B | $3.13 / 68.00% | $2.80 / 65.00% |
| Agentic Plan Caching (Ours) | GPT-4o | Llama-3.1-8B | Llama-3.1-8B | $1.86 / 85.50% | $2.03 / 82.00% |
| Agentic Plan Caching (Ours) | Claude 3.5 Sonnet | Llama-3.1-8B | Llama-3.1-8B | $2.56 / 88.00% | $2.73 / 81.50% |

Table 9: **Sensitivity Analysis of Large Planner LM: Results.**

| Method | Large Planner LM | Small Planner LM | Actor LM | FinanceBench | TabMWP |
|---|---|---|---|---|---|
| | | | | Cost↓ / Accuracy↑ | Cost↓ / Accuracy↑ |
| Full-History Caching | GPT-4o | Llama-3.1-8B | Llama-3.1-8B | $1.99 / 72.00% | $2.24 / 62.50% |
| Full-History Caching | GPT-4o | Qwen-2.5-7B | Llama-3.1-8B | $2.34 / 72.50% | $2.15 / 67.50% |
| Full-History Caching | GPT-4o | Llama-3.2-3B | Llama-3.1-8B | $1.93 / 67.00% | $1.67 / 56.00% |
| Agentic Plan Caching (Ours) | GPT-4o | Llama-3.1-8B | Llama-3.1-8B | $1.86 / 85.50% | $2.03 / 82.00% |
| Agentic Plan Caching (Ours) | GPT-4o | Qwen-2.5-7B | Llama-3.1-8B | $1.66 / 90.00% | $1.75 / 80.50% |
| Agentic Plan Caching (Ours) | GPT-4o | Llama-3.2-3B | Llama-3.1-8B | $1.62 / 84.00% | $1.88 / 80.00% |

Table 10: **Sensitivity Analysis of Small Planner LM: Results.**

| Method | Large Planner LM | Small Planner LM | Actor LM | FinanceBench | TabMWP |
|---|---|---|---|---|---|
| | | | | Cost↓ / Accuracy↑ | Cost↓ / Accuracy↑ |
| Accuracy-Optimal | GPT-4o | - | Llama-3.1-8B | $4.03 / 91.00% | $3.35 / 83.00% |
| Accuracy-Optimal | GPT-4o | - | Qwen-2.5-7B | $3.97 / 91.00% | $3.06 / 87.50% |
| Accuracy-Optimal | GPT-4o | - | Llama-3.2-3B | $4.16 / 81.50% | $4.43 / 74.00% |
| Cost-Optimal | Llama-3.1-8B | - | Llama-3.1-8B | $0.21 / 54.00% | $0.19 / 55.50% |
| Cost-Optimal | Llama-3.1-8B | - | Qwen-2.5-7B | $0.23 / 58.50% | $0.17 / 65.50% |
| Cost-Optimal | Llama-3.1-8B | - | Llama-3.2-3B | $0.17 / 53.00% | $0.24 / 57.00% |
| Full-History Caching | GPT-4o | Llama-3.1-8B | Llama-3.1-8B | $1.99 / 72.00% | $2.24 / 62.50% |
| Full-History Caching | GPT-4o | Llama-3.1-8B | Qwen-2.5-7B | $2.31 / 70.50% | $1.77 / 60.50% |
| Full-History Caching | GPT-4o | Llama-3.1-8B | Llama-3.2-3B | $2.20 / 60.00% | $2.22 / 58.00% |
| Agentic Plan Caching (Ours) | GPT-4o | Llama-3.1-8B | Llama-3.1-8B | $1.86 / 85.50% | $2.03 / 82.00% |
| Agentic Plan Caching (Ours) | GPT-4o | Llama-3.1-8B | Qwen-2.5-7B | $1.77 / 90.50% | $1.76 / 84.00% |
| Agentic Plan Caching (Ours) | GPT-4o | Llama-3.1-8B | Llama-3.2-3B | $1.85 / 80.00% | $2.18 / 73.00% |

Table 11: **Sensitivity Analysis of Actor LM: Results.**

# D   Example Agent Workflows and Cached Plan Templates

This section illustrates example agent workflows and their corresponding cached plan templates for data-intensive reasoning tasks. We begin with a simple single-turn task (more complex multi-turn examples are presented in our experiments):

> **Task:** What is FY2019 working capital ratio for Costco? Define working capital ratio as total current assets divided by total current liabilities. Round your answer to two decimal places. Give a response to the question by relying on the details shown in the statement of financial position.
> **Context:** {costco_financial_documents}

Below is the full agent workflow between a large planner LM and an actor LM, executed without caching (prompts omitted for brevity):

> **Planner:** 1. Decompose the Task: The task requires calculating the working capital ratio for Costco for FY2019. This involves two main components: identifying the total current assets and the total current liabilities for FY2019 from the statement of financial position.
> 2. Explain Each Component:
> - Identify Total Current Assets: This is important because it represents the resources that Costco can convert into cash within a year. Knowing this figure is essential for calculating the working capital ratio.
> - Identify Total Current Liabilities: This represents the obligations that Costco needs to settle within a year. This figure is crucial for determining the working capital ratio as it provides the denominator in the calculation.
> - Calculate Working Capital Ratio: This step involves dividing the total current assets by the total current liabilities. The result will give us the working capital ratio, which is a measure of Costco's short-term financial health.
> 3. Formulate a Focused Message: The next step is to ask the small language model for the specific figures of total current assets and total current liabilities for Costco in FY2019. This will allow us to perform the calculation.
> 4. Conclude with a Final Answer: Once we have the figures, we can calculate the working capital ratio and provide the final answer rounded to two decimal places.
> Generated plan (as a message to the actor LM): Please provide the total current assets and total current liabilities for Costco for FY2019 from the statement of financial position.
> **Actor:** Based on the provided statement of financial position for Costco Wholesale Corporation as of September 1, 2019, the total current assets and total current liabilities are:
> - Total current assets: $23,485 million
> - Total current liabilities: $23,237 million
> **Planner:** 1. What information we have gathered: We have obtained the total current assets and total current liabilities for Costco as of September 1, 2019. The total current assets are $23,485 million, and the total current liabilities are $23,237 million.
> 2. Whether it is sufficient to answer the question: Yes, this information is sufficient to calculate the working capital ratio for Costco for FY2019.
> 3. If not sufficient, what specific information is missing: Since we have both the total current assets and total current liabilities, no additional information is needed.
> 4. If sufficient, how we would calculate or derive the answer: To calculate the working capital ratio, we divide the total current assets by the total current liabilities. The formula is:
> Working Capital Ratio = Total Current Assets / Total Current Liabilities
> Substituting the given values: Working Capital Ratio = $23,485 million / $23,237 million
> Calculating this gives us approximately 1.01 when rounded to two decimal places.
> Conclusion: The working capital ratio for Costco for FY2019 is 1.01.

The following plan template is extracted from this successful workflow and stored in the cache:

> **Keyword:** working capital ratio
> **Plan Template:** 1. Plan (round 1): Retrieve total current assets and total current liabilities from the statement of financial position.
> 2. Expected response from actor LM (round 1): Total current assets and total current liabilities.
> 3. Plan (round 2): The working capital ratio can be determined by:
> Working Capital Ratio = Total Current Assets / Total Current Liabilities.

Later, for tasks with the same extracted keyword, such as the example below:

> **Task:** What is FY2021 working capital ratio for Best Buy? Define working capital ratio as total current assets divided by total current liabilities. Round your answer to two decimal places. Please base your judgments on the information provided primarily in the statement of financial position.
> **Context:** {bestbuy_financial_documents}

We reuse the cached plan template and adapt it using a small planner LM:

> **Planner:** (Using cached plan template) Please provide the total current assets and total current liabilities for Best Buy in FY2021 from the statement of financial position, so I can calculate the working capital ratio.
> **Actor:** According to the Consolidated Balance Sheets, the total current assets for Best Buy in FY2021 are $12,540 million, and the total current liabilities are $10,521 million.
> **Planner:** (Using cached plan template) The FY2021 working capital ratio for Best Buy is calculated by dividing the total current assets by total current liabilities. With total current assets of $12,540 million and total current liabilities of $10,521 million, the working capital ratio is $12,540 million / $10,521 million = 1.19. Therefore, the working capital ratio is 1.19, rounded to two decimal places.

Using the cached template significantly shortens the agent execution log, reducing token usage for expensive planner LMs. This efficiency gain comes from:

- Avoiding redundant planning for repeated tasks.
- Knowing when sufficient information has been gathered to terminate the workflow, thus avoiding unnecessary Plan-Act iterations.

## E   Extended Discussion of Limitations, Societal Impact, and Future Directions

**Limitations and Challenges**   First, we focus on two-stage Plan-Act agent architecture in this work. More complex multi-agent systems could present new challenges for maintaining cache consistency across multiple components. Second, for highly dynamic workloads with frequent task variations, the benefits of caching may diminish as historical plans may be less applicable. Finally, our evaluation primarily emphasizes cost reduction. Future work could consider additional system metrics such as latency, throughput, and computational overhead.

**Broader Impact and Societal Implications**   We believe that the proposed agentic plan caching framework has broader implications for AI accessibility and democratization. By reducing LLM serving costs, this framework could enable smaller enterprises, academic institutions, and individual developers to deploy agentic AI systems without incurring prohibitive API costs. Additionally, plan caches generated by advanced, commercial LLMs could potentially be shared or adapted for use with open-source models (as shown in our experiments), facilitating greater access to state-of-the-art agentic capabilities without direct reliance on expensive, closed-source APIs (*e.g.,* from OpenAI). This approach also raises questions about the long-term impact on data privacy, especially in cases where plan caches contain sensitive or proprietary information. Ensuring cache privacy and data security in LLM agents requires further research.

**Future Directions**   Several future directions could extend the utility of agentic plan caching. First, more advanced cache look-up and plan adaptation methods (like retrieval-augmented generation) might further enhance the relevance of cached plans in complex workflows. Second, enabling user-configurable cache parameters (*e.g.,* cache size, eviction strategies, fuzzy matching policies) could provide more control over caching strategies and allow for tailored cost-performance trade-offs. Finally, integrating the idea of agentic plan caching into existing LLM and agent serving frameworks at production scale would further enhance its applicability and impact. Overall, we hope this work inspires further research on optimizing the efficiency and cost-effectiveness of agentic AI systems.

