# OpenReview forum: "Agentic Plan Caching: Test-Time Memory for Fast and Cost-Efficient LLM Agents"
_NeurIPS.cc/2025/Conference — NeurIPS 2025 poster_

### Official Review · Reviewer_v2FN · 2025-06-30

**Clarity:** 3
**Significance:** 4
**Originality:** 3
**Rating:** 4
**Confidence:** 3

**Summary:**

Current LLM caching methods (e.g., context or semantic caching) fall short for agentic tasks due to dependencies on external data and dynamic environments. To address this, the authors introduce *agentic plan caching*, a novel technique that captures, stores, and reuses structured plan templates from past executions. By leveraging keyword-based matching and lightweight adaptation models, the system efficiently tailors cached plans to new tasks with similar semantics. Experiments across real-world applications demonstrate a 46.62% average cost reduction while maintaining performance, providing a scalable and cost-effective enhancement for LLM agent deployment. This approach complements existing LLM serving systems, bridging the gap for agent-specific optimization.

**Questions:**

1. In line 195-199, The Figure 4a does not show the Plan-Act Framework. Maybe it should be Figure 1a?
2. Could you include evaluation results from additional benchmarks?

**Ethical Concerns:**

["NO or VERY MINOR ethics concerns only"]

**Limitations:**

yes

**Quality:**

3

**Strengths And Weaknesses:**

**Strengths:**

- The paper is well-structured, with rigorous experiments and clear methodology. The evaluation on two datasets (FinanceBench and TabMWP) provides strong empirical support for the claims.
- The writing is clear, and the figures effectively illustrate the framework and results. The distinction between traditional caching and agentic plan caching is well-explained.
- The idea of caching plan templates for agentic workflows is novel and interesting, which can fills a gap left by existing caching techniques, offering practical cost savings without significant performance degradation.

**Weaknesses:**

- The evaluation is limited to two datasets and a specific agent architecture (planner-actor). Broader applicability to more complex multi-agent systems or diverse workflows remains unexplored.
- While cost reduction is the focus, other system metrics like latency and throughput are not discussed, which are important for real-world deployment.

---

> ### Author Rebuttal · Authors · 2025-07-31
>
> We thank the reviewer for the constructive and helpful feedback! We address each concern/question below.
>
> **#1: In line 195-199, The Figure 4a does not show the Plan-Act Framework. Maybe it should be Figure 1a?**
>
> Yes, this is a typo. It should be “Figure 1a”. Thank you so much for pointing it out! We will fix it in the final version of the paper.
>
> **#2: Could you include evaluation results from additional benchmarks?**
>
> We have added more experiment results to include tasks across diverse domains, including long-context reasoning, math problem-solving, etc. We are also actively working to incorporate benchmarks from domains like function calling and web agents. For these experiments, we use GPT-4o as the large planner LM, and Llama-3.1-8B-Instruct as the small planner LM and the actor LM. Below, we show one of the long-context benchmarks from the Minions paper:
>
> | Category | Benchmark | Method | Cost | Accuracy |
> |-------|-----|-------|-------|-------|
> | Long-context reasoning | QASPER (200 random samples) | Accuracy-Optimal | $2.14 | 58.00% |
> | | | Cost-Optimal | $0.21 | 53.00% |
> | | | **Ours** | **$0.78** | **57.00%** |
> | Math reasoning | AIME 2024 (all 30 queries) | Accuracy-Optimal | $1.14 | 64.52% |
> | | | Cost-Optimal | $0.65 | 48.39%  |
> | | | **Ours** | **$0.85** | **61.29%** |
> | Math reasoning | AIME 2025 (all 30 queries) | Accuracy-Optimal | $1.34 | 61.29% |
> | | | Cost-Optimal | $0.60 | 48.39% |
> | | | **Ours** | **$0.81** | **58.06%** |
>
> If you have a benchmark in mind, feel free to let us know!
>
> We have done a lot of additional in-depth microbenchmarks and analysis of our caching system to show how it could be adopted in practice, like scalability and online cache management, fuzzy keyword match v.s. exact keyword match, the cold-start effect of our system, etc. Due to space constraint, we won't be able to fit everything in this rebuttal response. Please feel free to take a look at our additional results under other reviews --- We are happy to answer any questions you may have about them!
>
> **#3: While cost reduction is the focus, other system metrics like latency and throughput are not discussed, which are important for real-world deployment.**
>
> We thank the authors for pointing out additional system metrics that are important in evaluation. We evaluated the wall-clock latency of our system and compared it to both accuracy-optimal and cost-optimal baselines. Additionally, we provide a detailed breakdown of latency incurred by each component in our plan caching pipeline. This microbenchmark is based on 100 randomly sampled queries from the FinanceBench dataset, with a cache hit rate of 46% (i.e., cached plans were used for 46 of the 100 queries). As shown in the table below, most of the additional latency in our system comes from LLM-powered cache generation, which takes an average of 3.99 seconds per entry. To further mitigate this overhead, we (1) automatically disable caching when the hit rate is consistently low, and (2) are exploring optimizations such as parallel cache generation and speculative next-query inference.
>
> | Method | Plan latency (s) | Act latency (s) | Keyword extraction latency (s) | Cache lookup latency (s) | Cache generation latency (s) | End-to-end latency (s) |
> |-------|-----|-------|-------|-------|-------|-------|
> | Accuracy-Optimal | 1813.41  |  94.39   | N/A | N/A | N/A | 1959.24 |
> | Cost-Optimal   |  856.75  |  93.31   | N/A | N/A | N/A | 1004.79 |
> | Ours | 1011.82 | 131.44 | 42.29 | < 1s | 215.80 | 1424.82 |
>
> Our system offers a favorable performance trade-off: We achieve significantly lower cost than the accuracy-optimal baseline while preserving high accuracy at the cost of moderate latency, most of which is attributable to one-time cache generation.
>
> We agree that throughput is an important system metric for LLM serving systems. We did not evaluate throughput in this work because agentic workloads usually do not emphasize batched inference, which is the primary driver of throughput in traditional LLM serving systems. Currently, the community’s focus is primarily on reducing API cost and improving latency, which our system directly targets. We see exploring throughput improvements, especially through plan caching and reuse in batched settings, as an exciting direction for future work.
>
> **#4: Broader applicability to more complex multi-agent systems or diverse workflows remains unexplored.**
>
> Thank you for raising this important point! We agree that agentic systems can exhibit a wide range of architectures and workflow patterns, making comprehensive coverage challenging in our evaluation. In this work, we focus on plan-act agents because this abstraction is widely adopted in both research and industry, e.g., ReAct and its variants, Minions, etc. Moreover, many complex multi-agent systems can be viewed as extensions of the plan-act paradigm, where planning and execution are distributed across agents. As a future direction, we plan to explore how plan caching can be generalized to support a broader range of agent architectures and workflows.
>
> **Finally:** We really appreciate your insightful suggestions, which have helped us strengthen the design and the evaluation sections of our work. If the response is helpful, we would be grateful if you could consider adjusting your score accordingly. We are committed to incorporating all promised additions and experiment results in the camera‑ready version to make the paper as valuable as possible to the community.

---

### Official Review · Reviewer_RMgA · 2025-07-01

**Clarity:** 3
**Significance:** 2
**Originality:** 3
**Rating:** 4
**Confidence:** 4

**Summary:**

The paper proposes agentic plan caching, a serving-time technique aimed at cutting the cost of Plan-Act LLM agents. After each completed run, the system extracts a plan template from the planner’s execution trace, stores it under a task-intent keyword, and later retrieves and lightly adapts that template—using a small model—when a semantically similar request arrives. Experiments on two data-intensive reasoning benchmarks (FinanceBench and Tabular-MWP) show a mean 46 % cost reduction while retaining 96 % of the accuracy of an all-large-LM baseline, with only ~1 % overhead for keyword extraction and cache management.

**Questions:**

1. Exact keyword matching avoids false positives but may miss near-duplicates. Have the authors explored lightweight embedding search or fuzzy matching, and how would that affect hit-rate, cost, and accuracy?

2. The paper focuses on token cost; could end-to-end latency numbers be reported (cache lookup + rewrite)?

**Ethical Concerns:**

["NO or VERY MINOR ethics concerns only"]

**Final Justification:**

I am mostly concern about the cache capacity and end-to-end latency. The authors have addressed almost all of my previous concerns, clarifying key methodological choices and strengthening the interpretation of the results. In light of these improvements, I am willing to raise my score.

**Limitations:**

See weakness.

**Quality:**

3

**Strengths And Weaknesses:**

Strength:

1. The paper addresses the rising serving cost of multi-step LLM agents, a practical bottleneck that is less studied than model-side optimization. The distinction between query-level (chatbot) caching and task-level (agent) caching is well argued.

2. The framework rests on off-the-shelf components and could be grafted onto other Plan-Act agents.

Weakness:

1. Scalability and cache management are not discussed. How does cost-saving scale with cache size? Is there a point of diminishing returns?

2. Cold-start and domain drift is not discussed. What happens when the cache is empty or when tasks drift over time? Could you report accuracy and cost as a function of cache warm-up and domain shift would clarify the real-world usefulness.

3. While cost is reduced, the extra keyword extraction, lookup, and template adaptation may add latency. Could the paper report end-to-end wall-clock times? In addition, storing plan templates might leak sensitive workflow information

4. Generality is not discussed. The solution is demonstrated with a specific large planner and an 8 B re-planner. How well does the approach transfer if those models are swapped for smaller or fundamentally different LLM families?

---

> ### Author Rebuttal · Authors · 2025-07-31
>
> We thank the reviewer for the constructive feedback!
>
> **#1: Scalability and cache management are not discussed.**
>
> Our microbenchmark reveals two key findings on the scalability of plan caching and cache management.
>
> (1) Larger cache, more cost and latency reduction (until the point of diminishing return): On the FinanceBench dataset, we find that a larger cache size would lead to larger reduction in end-to-end cost and latency, which is aligned with our intuition since a smaller cache size means more cost and latency dedicated to cache eviction and re-generating evicted cache entries. We present some numbers with a simple LRU eviction policy:
>
> | Cache size | Cache hit rate | Cost | Accuracy | Planning latency (s) | Cache generation latency (s) | End-to-end latency (s) |
> |-------|-----|-------|-------|-------|-------|-------|
> | 1 | 2% | $3.97 | 92.00% | 1638.85 | 383.87 | 2232.76 |
> | 10 | 13% | $3.51 | 88.00% | 1381.89 | 334.61 | 1911.95 |
> | 20 | 28% | $2.95 | 85.00% | 1248.06 | 289.36 | 1772.61 |
> | 50 | 45% | $1.88 | 86.00% | 1015.12 | 204.99 | 1459.92 |
> | 100 | 46% | $1.86 | 85.50% | 1011.81 | 215.80 | 1424.82 |
>
> **Will there be a point of diminishing return?** Yes, when the cache size is large enough, e.g. larger than the unique number of task intents in the entire agentic dataset or workload, there will be no extra benefit of having a larger cache. In practice, users could flexibly adjust cache size cap based on speed and storage requirements.
>
> (2) Scalability of cache operations (lookup and insertion): For exact-match cache lookup, our cache uses Python’s native dictionary implementation, which is highly optimized. Both lookup and insertion have an average-case time complexity of O(1). To empirically validate the performance, we measured wall-clock latency for cache hits and misses with increasing cache sizes. We cleared the CPU cache before each run and averaged each measurement over 100 trials. For exact match (used in our experiments), lookup latency remains low up to a cache size of 10^6
>
> | Cache size | Avg. cache hit latency (μs) | Avg. cache miss latency (μs) |
> |-------|-----|-------|
> | 10^2 | 13 | 14 |
> | 10^3 | 15 | 15 |
> | 10^4 | 16 | 17 |
> | 10^5 | 22 | 18 |
> | 10^6 | 56 | 37 |
>
> In contrast, fuzzy matching usually incurs significantly higher latency (depending on implementation detail of the fuzzy match algorithm), which grows rapidly with cache size. We implemented a semantic similarity-based fuzzy cache using ```SentenceTransformer('all-MiniLM-L6-v2')``` from the ```sentence_transformers``` library, with a similarity threshold of 0.8 for hit detection:
>
> | Cache size | Avg. cache hit latency (μs) | Avg. cache miss latency (μs) |
> |-------|-----|-------|
> | 10^2 | 57 | 24 |
> | 10^3 | 75 | 70 |
> | 10^4 | 581 | 554 |
> | 10^5 | 10388 | 10317 |
> | 10^6 | 148449 | 148147 |
>
> These results confirm that fuzzy matching introduces substantial overhead and scales poorly compared to exact match. This is one of the key reasons we adopt exact keyword matching in our system, as discussed in Section 3.2. Additional factors include the difficulty of selecting a universal similarity threshold, which is a common challenge in semantic caching.
>
> Our cache provides a reconfigurable API that allows users to customize key settings, such as maximum cache size, eviction policy, and lookup rules (e.g., fuzzy keyword matching), to suit the needs of specific agentic applications. We will include a more detailed discussion of this API in the final version of the paper.
>
> **#2: Cold-start and domain drift is not discussed.**
>
> Thank you for this important point. Cold-start is an inherent limitation of test-time plan caching (in contrast to offline plan caching), as the cache starts empty. During the early stages, the system incurs additional latency and cost from generating new cache entries. To study this, we conduct a time-series analysis of cache warm-up:
>
> | Query percentile | # cache entries | Cache hit rate | Cost | Planning latency (s) | Cache generation latency (s) | E2E Latency (s) |
> |-------|-----|-------|-------|-------|-------|-------|
> | 20th | 15 | 14.29% | $0.59 **(32.07%)** | 260.19 **(27.43%)** | 59.19 **(27.64%)** | 358.71 **(25.72%)** |
> | 40th | 27 | 24.39% | $0.97 **(52.72%)** | 484.88 **(51.12%)** | 130.29 **(60.86%)** | 689.49 **(49.44%)** |
> | 60th | 36 | 36.07% | $1.20 **(65.22%)** | 638.12 **(67.28%)** | 167.89 **(78.41%)** | 926.82 **(66.46%)** |
> | 80th | 42 | 40.75% | $1.63 **(88.59%)** | 820.94 **(86.56%)** | 195.59 **(91.35%)** | 1183.39 **(84.86%)** |
> | 100th | 46 | 48.00% | $1.84 **(100.00%)** | 948.36 **(100.00%)** | 214.11 **(100.00%)** | 1394.56 **(100.00%)** |
>
> As the cache grows, hit rate increases, and marginal latency and cost decrease over time. In practice, if the target workload is known ahead of time, users can proactively warm up the cache using offline samples before deployment to mitigate cold-start effects.
>
> Regarding domain drift, we agree this is a practical concern for any caching-based system. Cached entries may become stale as task distributions shift. To address this, our system supports configurable cache invalidation policies (e.g., time-based expiration), allowing users to tailor freshness guarantees to their applications. While caching is most effective when domain drift is limited, we will include further discussion on how to mitigate drift in the final version.
>
> **#3: Storing plan templates might leak sensitive workflow information.**
>
> We agree that security and privacy are important considerations when caching plans, especially when templates are shared across agents from different users. To mitigate this risk, we do not cache raw execution histories. Instead, we extract plan templates by filtering out query-specific details such as user or company names. We hope our work encourages further research on privacy-preserving agent systems and memory mechanisms.
>
> **#4: Generality is not discussed. The solution is demonstrated with a specific large planner and an 8 B re-planner. How well does the approach transfer if those models are swapped for smaller or fundamentally different LLM families?**
>
> Thank you for raising this point! We address generality in Appendix C, where we conduct a sensitivity analysis using a range of commercial and open-source models as the large planner LM, the small planner LM, and actor LMs. Our results show that the system consistently improves performance across different model families (e.g., OpenAI, Claude, LLaMA, Qwen) and sizes.
>
> **#5: Exact keyword matching avoids false positives but may miss near-duplicates. Have the authors explored lightweight embedding search or fuzzy matching, and how would that affect hit-rate, cost, and accuracy?**
>
> We have implemented fuzzy keyword matching in our prototype, and we present some results below. Specifically, we use semantic embeddings to measure similarity between keywords, treating a cache lookup as a hit if the similarity exceeds a predefined threshold. Below, we report the empirical results:
>
> | Similarity threshold | Cache hit rate | Cost | Accuracy | Planning latency (s) | End-to-end latency (s) |
> |-------|-----|-------|-------|-------|-------|
> | =100% | 46% | $1.86 | 85.50% | 1011.82 | 1424.82 |
> | >80% | 54% | $1.15 | 83.00% | 875.31 | 1219.73 |
> | >60% | 64% | $0.93 | 77.00% | 720.29 | 1044.50 |
>
> As we can observe, lowering the similarity threshold increases the cache hit rate and reduces cost and latency, but at the expense of accuracy. This highlights the inherent trade-off in fuzzy matching: While more aggressive matching improves efficiency, it risks introducing less relevant cached plans.
>
> The good news is that our cache interface is highly reconfigurable. Users can flexibly enable fuzzy matching and tune the similarity threshold to fit their application's tolerance for semantic drift and performance requirements.
>
> **#6: The paper focuses on token cost; could end-to-end latency numbers be reported (cache lookup + rewrite)?**
>
> We thank the reviewer for pointing out that latency is an important metric in agent system performance measurement! We evaluated the wall-clock latency of our system and compared it to both accuracy-optimal and cost-optimal baselines. Additionally, we provide a detailed breakdown of latency incurred by each component in our plan caching pipeline. This microbenchmark is based on 100 randomly sampled queries from the FinanceBench dataset, with a cache hit rate of 46% (i.e., cached plans were used for 46 of the 100 queries). As shown in the table below, most of the additional latency in our system comes from LLM-powered cache generation, which takes an average of 3.99 seconds per entry. To further mitigate this overhead, we (1) automatically disable caching when the hit rate is consistently low, and (2) are exploring optimizations such as parallel cache generation and speculative next-query inference.
>
> | Method | Plan latency (s) | Act latency (s) | Keyword extraction latency (s) | Cache lookup latency (s) | Cache generation latency (s) | End-to-end latency (s) |
> |-------|-----|-------|-------|-------|-------|-------|
> | Accuracy-Optimal | 1813.41  |  94.39   | N/A | N/A | N/A | 1959.24 |
> | Cost-Optimal   |  856.75  |  93.31   | N/A | N/A | N/A | 1004.79 |
> | Ours | 1011.82 | 131.44 | 42.29 | < 1s | 215.80 | 1424.82 |
>
> Our system offers a favorable performance trade-off: We achieve significantly lower cost than the accuracy-optimal baseline while preserving high accuracy at the cost of moderate latency, most of which is attributable to one-time cache generation.
>
> **Finally:** We really appreciate your insightful suggestions, which have helped us strengthen the design and the evaluation sections of our work. If the response is helpful, we would be grateful if you could consider adjusting your score accordingly. We are committed to incorporating all promised additions and experiment results in the camera‑ready version to make the paper as valuable as possible to the community.

---

> > ### Comment · Reviewer_RMgA · 2025-08-04
> > **Ack**
> >
> > The authors have addressed almost all of my previous concerns, clarifying key methodological choices and strengthening the interpretation of the results. In light of these improvements, I am willing to raise my score.

---

### Official Review · Reviewer_c5yq · 2025-07-03

**Clarity:** 4
**Significance:** 2
**Originality:** 3
**Rating:** 5
**Confidence:** 4

**Summary:**

This paper proposes an agentic plan caching algorithm that reduces costs by nearly 50% and nearly maintains the original optimal accuracy. Previous caching methods are limited because they (1) have model-specific constraints; agentic flows typically use various models while caching methods are for specific models (2) reuse cached responses while agentic flows have data-dependent outputs (3) are limited in adaptability sometimes requiring exact text matches. The agentic plan caching framework, intended for use in Plan-Act frameworks that utilize a large LM for high level planning and a small LM to carry out the plan, generates a plan template on cache misses to be followed by a small LM during cache hits. Evaluation is conducted on methods optimal for accuracy and cost and semantic caching and agentic plan caching is shown to reach the best tradeoff between cost and accuracy.

**Questions:**

- On the discussion between exact matching versus fuzzy matching, an argument is made that fuzzy matching could make cache lookups slower and costly. An empirical evaluation of this claim would make this point stronger.
  - Additionally, there is no study of the speed of the agentic plan caching algorithm. Surely given a plan template is generated this would be much slower than other approaches.
- What is the major bottleneck in keeping the costs low but the accuracy high? The agentic plan caching is nearly close to optimal in some cases (Table 1 TabMWP) which is impressive but what would close this gap further?
  - I am interested in more information regarding the rule-based filter and specific failure modes of the plan template that lead to unexpected behavior and the lower performance

**Ethical Concerns:**

["NO or VERY MINOR ethics concerns only"]

**Final Justification:**

My initial concerns were regarding latency of the overall system as well as understanding where the bottleneck in the system lies. Comprehensive evaluations were introduced that fully addressed all my concerns regarding latency and I understand the bottleneck to be the capabilities of the small planner LM. I have decided to increase my score.

**Limitations:**

The authors have clearly addressed the limitations of this work.

**Quality:**

4

**Strengths And Weaknesses:**

**Quality:** The baselines and experiments effectively demonstrate the usefulness of agentic plan caching.

**Clarity:** The paper is written clearly, with the shortcomings of previous caching approaches clearly demonstrated.

**Significance:** The cost-effectiveness of this caching strategy is very significant but is limited to two-stage Plan-Act agent architectures whereas other caching strategies can be applied agnostic to architectures.

**Originality:** This caching approach is a unique addition to previous approaches like KV caching and semantic caching.

---

> ### Author Rebuttal · Authors · 2025-07-31
>
> We thank the reviewer for the kind words and constructive feedback! Below, we address each concern in detail.
>
> **#1: On the discussion between exact matching versus fuzzy matching, an argument is made that fuzzy matching could make cache lookups slower and costly. An empirical evaluation of this claim would make this point stronger.**
>
> Thank you for this insightful question. For exact-match cache lookup, our cache uses Python’s native dictionary implementation, which is highly optimized. Both lookup and insertion have an average-case time complexity of O(1). To empirically validate the performance, we measured wall-clock latency for cache hits and misses with increasing cache sizes. We cleared the CPU cache before each run and averaged each measurement over 100 trials. For exact match (used in our experiments), lookup latency remains low up to a cache size of 10^6:
>
> | Cache size | Avg. cache hit latency (μs) | Avg. cache miss latency (μs) |
> |----|-------|-------|
> | 10^2 | 13 | 14 |
> | 10^3 | 15 | 15 |
> | 10^4 | 16 | 17 |
> | 10^5 | 22 | 18 |
> | 10^6 | 56 | 37 |
>
> In contrast, fuzzy matching usually incurs significantly higher latency (depending on implementation detail of the fuzzy match algorithm), which grows rapidly with cache size. We implemented a semantic similarity-based fuzzy cache using ```SentenceTransformer('all-MiniLM-L6-v2')``` from the ```sentence_transformers``` library, with a similarity threshold of 0.8 for hit detection:
>
> | Cache size | Avg. cache hit latency (μs) | Avg. cache miss latency (μs) |
> |-------|-----|-------|
> | 10^2 | 57 | 24 |
> | 10^3 | 75 | 70 |
> | 10^4 | 581 | 554 |
> | 10^5 | 10388 | 10317 |
> | 10^6 | 148449 | 148147 |
>
> These results confirm that fuzzy matching introduces substantial overhead and scales poorly compared to exact match. This is one of the key reasons we adopt exact keyword matching in our system, as discussed in Section 3.2. Additional factors include the difficulty of selecting a universal similarity threshold, which is a common challenge in semantic caching.
>
> To further investigate, we implemented fuzzy keyword matching in our prototype. Specifically, we use semantic embeddings to measure similarity between keywords, treating a cache lookup as a hit if the similarity exceeds a predefined threshold. Below, we report the empirical results:
>
> | Similarity threshold | Cache hit rate | Cost | Accuracy | Planning latency (s) | End-to-end latency (s) |
> |-------|-----|-------|-------|-------|-------|
> | =100% | 46% | $1.86 | 85.50% | 1011.82 | 1424.82 |
> | >80% | 54% | $1.15 | 83.00% | 875.31 | 1219.73 |
> | >60% | 64% | $0.93 | 77.00% | 720.29 | 1044.50 |
>
> As we can observe, lowering the similarity threshold increases the cache hit rate and reduces cost and latency, but at the expense of accuracy. This highlights the inherent trade-off in fuzzy matching: While more aggressive matching improves efficiency, it risks introducing less relevant cached plans.
>
> The good news is that our cache interface is highly reconfigurable. Users can flexibly enable fuzzy matching and tune the similarity threshold to fit their application's tolerance for semantic drift and performance requirements.
>
> **#2: Speed of the proposed system?**
>
> We evaluated the wall-clock latency of our system and compared it to both accuracy-optimal and cost-optimal baselines. Additionally, we provide a detailed breakdown of latency incurred by each component in our plan caching pipeline. This microbenchmark is based on 100 randomly sampled queries from the FinanceBench dataset, with a cache hit rate of 46% (i.e., cached plans were used for 46 of the 100 queries). As shown in the table below, most of the additional latency in our system comes from LLM-powered cache generation, which takes an average of 3.99 seconds per entry. To further mitigate this overhead, we (1) automatically disable caching when the hit rate is consistently low, and (2) are exploring optimizations such as parallel cache generation and speculative next-query inference.
>
> | Method | Plan latency (s) | Act latency (s) | Keyword extraction latency (s) | Cache lookup latency (s) | Cache generation latency (s) | End-to-end latency (s) |
> |-------|-----|-------|-------|-------|-------|-------|
> | Accuracy-Optimal | 1813.41  |  94.39   | N/A | N/A | N/A | 1959.24 |
> | Cost-Optimal   |  856.75  |  93.31   | N/A | N/A | N/A | 1004.79 |
> | Ours | 1011.82 | 131.44 | 42.29 | < 1s | 215.80 | 1424.82 |
>
> Our system offers a favorable performance trade-off: We achieve significantly lower cost than the accuracy-optimal baseline while preserving high accuracy at the cost of moderate latency, most of which is attributable to one-time cache generation.
>
> **#3: Major bottleneck in keeping the costs low but the accuracy high. Specific failure modes of the plan template that lead to unexpected behavior and the lower performance.**
>
> For FinanceBench, a key challenge arises from the limited capacity of the small planner LM, which must not only follow the plan but also aggregate context across multiple agentic rounds. This includes retrieving relevant information from prior rounds and actor LMs, then synthesizing a final answer, often requiring arithmetic or simple reasoning. We find that even when the correct plan is generated from the template and all necessary information is collected, the small planner LM may still fail to produce the correct final answer due to issues such as hallucinations or arithmetic mistakes. Enhancing the small planner LM’s capability in long-horizon reasoning remains an important direction for future work.
>
> A second failure mode stems from the need for robustness when actor LMs return invalid outputs or errors. In such cases, the planner LM must re-execute earlier steps before proceeding (“retry”). While our template serves as an in-context guide, enabling the planner to handle such deviations, we observe that small planner LMs, especially those under 8B, sometimes fail to confirm successful completion of earlier steps before moving forward. This can lead to cascading errors. We believe that adding stronger runtime checks would help ensure that the planner LM adapts to feedback from the actor LMs and environment, rather than rigidly following the cached plan.
>
> **Finally:** We really appreciate your insightful suggestions, which have helped us strengthen the design and the evaluation sections of our work. If the response is helpful, we would be grateful if you could consider adjusting your score accordingly. We are committed to incorporating all promised additions and experiment results in the camera‑ready version to make the paper as valuable as possible to the community.

---

> > ### Comment · Reviewer_c5yq · 2025-08-07
> >
> > Thank you for the insightful response. Fuzzy matching is slower from a retrieval standpoint but when putting it in the framework it can speed up the framework by increasing cache hits but at the cost of accuracy. The plan latency of your system is also not as slow as I expected (only 18% slower than the cost-optimal). Finally, the bottleneck is just a tradeoff with the small planner LM, as choosing smaller models keeps costs and inference time low but choosing larger ones improves the accuracy. My concerns have been addressed so I will be increasing my score.

---

### Official Review · Reviewer_AHQG · 2025-07-03

**Clarity:** 3
**Significance:** 3
**Originality:** 3
**Rating:** 4
**Confidence:** 3

**Summary:**

This paper proposes a cost-efficient serving strategy for LLM-based agents through plan caching. Instead of relying on conventional context or semantic caching, the authors introduce a method that caches structured plan templates extracted from past executions. New tasks are matched to cached plans using keyword-based similarity, and a smaller LLM is used to adapt these plans to the current context. Experiments on two agentic benchmarks demonstrate that this approach reduces serving costs significantly (by ~46%) while maintaining high task performance.

**Questions:**

Please see Weaknesses above.

**Ethical Concerns:**

["NO or VERY MINOR ethics concerns only"]

**Final Justification:**

The paper is well-written with good empirical results. Most of my concerns were addressed during the rebuttal. Therefore, I am maintaining my score.

**Limitations:**

Yes

**Quality:**

3

**Strengths And Weaknesses:**

## Strengths:
- The paper is overall well-written and includes clear, helpful figures.

- The analysis is thorough, especially in justifying design choices such as using keyword-based cache search instead of query-based methods, and showing the benefit of plan caching over semantic or full-history caching.

## Weaknesses:

- The TabMWP dataset feels a bit too simple. It would make the paper more impactful to evaluate on more complex math datasets like AIME.

- There is a lack of analysis on how the caching method performs as task diversity increases. For example, how would it perform on a benchmark with very diverse tasks, such as GAIA? Would it still outperform other caching strategies?

---

> ### Author Rebuttal · Authors · 2025-07-31
>
> We thank the reviewer for the kind words and constructive feedback! Below, we address each concern in detail.
>
> **#1: TabMWP is too simple, need to evaluate on more complex math datasets like AIME.**
>
> We have added results on AIME 2024 and AIME 2025, each consisting of 30 math reasoning problems. Although our system was not specifically designed for math tasks, we find that caching plans in the form of meta problem-solving strategies (such as reusable tricks and heuristics) significantly boosts performance of smaller reasoning LMs on new problems within similar domains (e.g., algebra, geometry). For these experiments, we use o3 as the large planner LM, and o3-mini as the small planner LM and the actor LM, since reasoning capability is quite important in these benchmarks.
>
> | Category | Benchmark | Method | Cost | Accuracy |
> |-------|-----|-------|-------|-------|
> | Math reasoning | AIME 2024 (all 30 queries) | Accuracy-Optimal | $1.14 | 64.52% |
> | | | Cost-Optimal | $0.65 | 48.39%  |
> | | | Ours | $0.85 | 61.29% |
> | Math reasoning | AIME 2025 (all 30 queries) | Accuracy-Optimal | $1.34 | 61.29% |
> | | | Cost-Optimal | $0.60 | 48.39% |
> | | | Ours | $0.81 | 58.06% |
>
> **#2: Potential lack of analysis on increased task diversity, e.g. on GAIA.**
>
> We have added more experiment results to include tasks across diverse domains, including long-context reasoning, math problem-solving, etc. We are also actively working to incorporate benchmarks from domains like function calling and web agents. For these experiments, we use GPT-4o as the large planner LM, and Llama-3.1-8B-Instruct as the small planner LM and the actor LM. Below, we show one of the long-context benchmarks from the Minions paper:
>
> | Category | Benchmark | Method | Cost | Accuracy |
> |-------|-----|-------|-------|-------|
> | Long-context reasoning | QASPER (200 random samples) | Accuracy-Optimal | $2.14 | 58.00% |
> | | | Cost-Optimal | $0.21 | 53.00% |
> | | | **Ours** | **$0.78** | **57.00%** |
>
> While our approach to plan caching is generally applicable to agentic tasks that benefit from reuse of reasoning patterns, we have not included GAIA due to its highly heterogeneous task queries, which limit opportunities for cache reuse. We will include a discussion in the camera-ready version on domains that benefit most from our method, as well as those where impact may be limited.
>
> If you have more benchmarks in mind, feel free to let us know!
>
> We have done a lot of additional in-depth microbenchmarks and analysis of our caching system to show how it could be adopted in practice, like scalability and online cache management, fuzzy keyword match v.s. exact keyword match, end-to-end latency analysis and breakdown, the cold-start effect of our system, etc. Due to space constraint, we won't be able to fit everything in this rebuttal response. Please feel free to take a look at our additional results under other reviews --- We are happy to answer any questions you may have about them!
>
> **Finally:** We really appreciate your insightful suggestions, which have helped us strengthen the evaluation part of our work. If the response is helpful, we would be grateful if you could consider adjusting your score accordingly. We are committed to incorporating all promised additions and experiment results in the camera‑ready version to make the paper as valuable as possible to the community.

---

> ### Comment · Reviewer_AHQG · 2025-08-08
>
> I thank the authors for their detailed response. I think the paper is in good shape and it would be strengthened further with a thorough analysis of the trade-off between cache hit rate and the similarity between tasks. I believe the hit rate should be worse as the tasks become more diverse; this is expected and is not considered a weak point. However, it would be useful to see at what point the tasks are too diverse for this framework to remain effective. That said, I thank the authors again, and I will maintain my score.

---

### Official Review · Reviewer_7A5h · 2025-07-05

**Clarity:** 3
**Significance:** 2
**Originality:** 2
**Rating:** 4
**Confidence:** 4

**Summary:**

This work proposes a new technique to do plan caching. Previous works have done caching to save KV parameters of the LLM, or the whole task including all parameters and keywords. The novelty of this approach is that the abstract plan or “plan template” is cached. The keywords are abstracted or removed before doing the caching. This makes the plan applicable for a wide range of parameters, and not just the single task it was originally meant for.

The advantage of the approach is that it saves the cost of using LLMs for plan generation every time, many of which are usually unnecessary. Experimental comparison with other caching techniques show that the cost is substantially reduced in two domains with little loss of plan quality.

**Questions:**

1. When there are multiple possible items in the cache matching the new task, how are they ranked? Have you encountered any such cases?
2. In Figure 1, why is there no link from the context to the planner?
3. In 2.2 (1) Model Specific constraints, why would one want to use a different LLM agent for a task of one agent?

**Ethical Concerns:**

["NO or VERY MINOR ethics concerns only"]

**Final Justification:**

The authors have clarified my doubts about the originality compared to classical case based planning literature, and promised to include a discussion of it in the revised version. They have also answered my questions related to the approach convincingly. In light of that, I am increasing my score.

**Limitations:**

Yes

**Paper Formatting Concerns:**

No concerns.

**Quality:**

2

**Strengths And Weaknesses:**

The paper is generally well written and easy to follow. The paper is also well motivated. The existing caching techniques in LLMs are explained well with figures.

The proposed method is relatively straighforward and makes sense. The concept of saving the abstract plans is not new and has been explored extensively in the case based reasoning and case based planning literature.

Borrajo, Daniel, Anna Roubíčková, and Ivan Serina. "Progress in case-based planning." ACM Computing Surveys (CSUR) 47, no. 2 (2015): 1-39.

Spalzzi, Luca. "A survey on case-based planning." *Artificial Intelligence Review* 16 (2001): 3-36.

Bergmann, Ralph, and Wolfgang Wilke. "On the role of abstraction in case-based reasoning." In *European Workshop on Advances in Case-Based Reasoning*, pp. 28-43. Berlin, Heidelberg: Springer Berlin Heidelberg, 1996.

Given the simplicity of the technique, I was expecting a deeper dive into the issues of abstraction and I was also expecting a more elaborate experimental evaluation.

When there are multiple possible items in the cache matching the new task, how are they ranked? Have you encountered any such cases?

A planner usually takes in the current context as a part of the initial state. So, why is there no link from the context to the Planner LM in the Plan-Act flow chart?

The main weakness I see is the lack of experimental domains and analysis.

---

> ### Author Rebuttal · Authors · 2025-07-31
>
> We thank the reviewer for the kind words and helpful feedback! We address each concern/question below.
>
> **#1: The concept of saving abstract plans is not new and has been explored extensively in the case‑based reasoning / planning (CBP) literature.**
>
> Although our work indeed shares the high‑level intuition of “store & reuse”, it targets a fundamentally different setting than classic symbolic CBP: LLM‑based, neural agents executing open‑ended natural‑language actions. In particular, our system differs from CBP in two key ways:
> (1) Automatic plan template extraction at test time from unconstrained and open-ended LLM generations, instead of storing hand‑crafted symbolic plans, and
> (2) Leveraging plan storage and reuse for cost-efficient inference of LLM agents.
> We will add an explicit discussion contrasting these ideas and citing the survey papers you helpfully provided (Borrajo et al., Spalzzi 2001, Bergmann & Wilke 1996).
>
> **#2: Depth of abstraction discussion, and potential lack of experimental domains / analysis.**
>
> Section 3.2 motivates some key design choices behind the simple abstraction of “store & reuse”, like why we abstract executions into templates and why simple keyword keys minimize false positives / negatives.
>
> We have done a lot of additional in-depth microbenchmarks and analysis of our caching system to show how it could be adopted in practice, like scalability and online cache management, fuzzy keyword match v.s. exact keyword match, end-to-end latency analysis and breakdown, the cold-start effect of our system, etc. Due to space constraint, we won't be able to fit everything in this rebuttal response. Please feel free to take a look at our additional results under other reviews --- We are happy to answer any questions you may have about them!
>
> We have also added experiments that apply our system to a wider set of tasks: long‑context reasoning and math reasoning, and we are also working to incorporate more benchmarks in agentic function calling and tool use. The results show that our plan‑act approach generalizes well across these domains, delivering lower cost while preserving accuracy. In the camera‑ready version, we will also discuss additional domains that could benefit most from our system and those where its impact may be limited.
>
> | Category | Benchmark | Method | Cost | Accuracy |
> |-------|-----|-------|-------|-------|
> | Long-context reasoning | QASPER (200 random samples) | Accuracy-Optimal | $2.14 | 58.00% |
> | | | Cost-Optimal | $0.21 | 53.00% |
> | | | **Ours** | **$0.78** | **57.00%** |
> | Math reasoning | AIME 2024 (all 30 queries) | Accuracy-Optimal | $1.14 | 64.52% |
> | | | Cost-Optimal | $0.65 | 48.39%  |
> | | | **Ours** | **$0.85** | **61.29%** |
> | Math reasoning | AIME 2025 (all 30 queries) | Accuracy-Optimal | $1.34 | 61.29% |
> | | | Cost-Optimal | $0.60 | 48.39% |
> | | | **Ours** | **$0.81** | **58.06%** |
>
> **#3: When there are multiple possible items in the cache matching the new task, how are they ranked? Have you encountered any such cases?**
>
> We have encountered such cases, especially when agentic queries share similar task intent. Our cache is highly configurable: users can choose to store multiple entries per task keyword or just one. When multiple entries are kept, any or all of them can be provided to the lightweight Planner LM, similar to multi-shot in-context learning. Users may also implement ranking strategies, e.g., ranking entries by recency might help handle domain shifts. In our long-context reasoning experiments, we keep a single entry per task keyword, as extracted task templates tend to be similar across shared intents. This may not hold in other applications.
>
> **#4: In Figure 1, why is there no link from the context to the planner?**
>
> We agree that some metadata about the Context may be useful for the Planner LM. However, we view this as an implementation detail that would unnecessarily clutter the diagram and is not essential for the common tasks we consider. For instance, to answer the query “What is FY2019 working capital ratio for Costco?”, the Planner does not need to read the entire 10-K to devise a strategy for answering the question. Still, it may be helpful for the Planner to know in advance that the Context is a 10-K report and have general knowledge of what such reports typically contain. We will adjust the diagram accordingly in the camera-ready version, e.g. by adding a dashed line between Context and Planner.
>
> **#5: In 2.2 (1) Model Specific constraints, why would one want to use a different LLM agent for a task of one agent?**
>
> Even for single-agent tasks, users may choose different base models based on performance goals. For example, OpenAI o3 for stronger reasoning, GPT-4.1 for better long-context handling, or 4o-mini for cost efficiency. The main challenge for KV cache sharing is that it remains difficult to transfer knowledge between agents with the same architecture but different base models.
>
> **Finally:** We really appreciate your insightful suggestions, which have already helped us strengthen both the analysis and presentation of this work. We hope the above clarifications resolve the concerns. If the response is helpful, we would be grateful if you could consider adjusting your score accordingly. We are committed to incorporating all promised additions in the camera‑ready version to make the paper as valuable as possible to the community.

---

### Note · Authors · 2025-08-16

We sincerely thank the AC and the reviewers for constructive feedback! We summarize the key takeaways from the discussion:

## Summary and Key Contributions

We propose **Agentic Plan Caching (APC)**: After an agent correctly completes a workflow, we extract a reusable plan template from its execution trace. For future semantically similar queries, we retrieve and adapt the template using a small model instead of replanning with a large one. APC targets Plan–Act agents, where planning dominates token cost.

Reviewers recognized APC as a novel and useful system for agent caching. Highlights:
- **Strong accuracy-cost trade-off**: ~46.6% cost reduction with ~96.7% of the accuracy of the optimal baseline.
- **Task-level caching**: Structured plans adapt to instance-specific context and handle data-dependent actions better than query/context-level caching.
- **Low overhead**: Cache construction adds only ~1.04% to total cost.
- **Model-agnostic**: Sensitivity studies show consistent gains across diverse LMs.

## Reviewer Concerns and Planned Revisions

- **Case-Based Planning (CBP) [7A5h].** We will add a paragraph contrasting APC with CBP and citing the surveys suggested.
- **Latency and Systems Analysis [c5yq, RMgA, v2FN].** We now report end-to-end latency with breakdown analysis. Results show APC preserves or improves latency via lightweight planning; cache overhead is minimal and auto-disabled under low hit rates.
- **Exact vs. Fuzzy Retrieval [c5yq, RMgA].** We added experiments on hit-rate vs. overhead trade-offs. We recommend exact-match by default, with fuzzy retrieval as an option for noisy queries, along with threshold guidance.
- **Cold Start, Domain Drift, and Cache Management [RMgA].** We report latency at different cache sizes and describe support for configurable strategies (e.g., eviction). A time-series analysis of cache warm-up is included.
- **Scope and Generality [7A5h, AHQG, v2FN].** We added new benchmarks (e.g., QASPER, AIME 2024/2025) and discuss when reuse chances diminish (e.g., GAIA). Model generalization results (across families/sizes) will be mentioned in the main text.
- **Other Clarifications [7A5h, v2FN].** We will discuss multi-template ranking (e.g., recency, past success), clarify multi-shot adaptation, fix the Figure 1 reference, and add a dashed Context→Planner edge.

We thank the AC and reviewers again for the engaging discussion! We will incorporate these revisions to strengthen clarity, completeness, and impact.

---

### Decision · Program_Chairs · 2025-09-17

**Decision:**

Accept (poster)

**Comment:**

# Summary

This paper introduces a caching strategy aimed at enhancing the cost‑effectiveness of employing large language models (LLMs) as agents for complex tasks. The core concept involves **extracting a plan template** from an execution log and storing it in a cache keyed by a **specific keyword**. For subsequent queries, the system first identifies a relevant keyword; if the cache contains a matching entry, the cached plan template is combined with a smaller, more economical LLM to generate the response. If the cache does not contain a suitable entry, the system falls back to using a larger, more expensive LLM to carry out the task.

The two major components in the proposed caching approach are:
1. Extracting Keyword: An LLM is used for identifying the keyword t
2. Extracting plan-template:  First action strings are extracted from execution log using a rule-based approach and then LLM is used to remove all the problem-specific details from those action strings.

Their experiments on two benchmarks---FinanceBench and Tabular‑MWP---show an average 46% cost reduction while retaining 96 % of the accuracy of an all-large-LM baseline, with only ~1 % overhead for keyword extraction and cache management. During rebuttal, the authors augment the evaluation with additional domains---long‑context QASPER and math AIME---and provide extensive micro‑benchmarks on latency, cache scalability, fuzzy vs. exact matching, cold‑start, and domain drift.



# Strengths

- **Novel**: The problem focused by the paper is novel and would be increasingly important to address as more agentic solutions are adopted. Definetly something that the NeurIPS community would be interested in.
- **Practical**: Proposed solution is straight forward, practical and adaptable. It uses off-the-shelf components at test-time and do not require a lot of investment/compute to adapt to different problems/domains.
- **Significant Empirical Evalutions**: The experiment results are strong and addition of experiments during the rebuttal make them even stronger.



# Weakness

- **Lack of Alternative Exploration**: The main concern I have is that the paper lacks an in-depth analysis of alternatives that could have been used for keyword extraction and plan-template extraction (or plan abstraction).  (Reviewer 7A5h)
- **Missing Failure Analysis**: It is unclear whether the failures stemmed from the actor LM, from false-positive cache hits, or from incorrect keyword matches. While the rebuttal offered some insight---suggesting that smaller LMs struggle with long‑context reasoning and may generate erroneous actions—a more comprehensive, quantitative analysis is still missing. (Reviewer c5yq)
- **Implications for Benchmarks that require Planning**: Finally, my last concern is about the utility of the proposed approach for benchmarks that specifically require planning. The benchmarks that are used for the analysis are complex reasoning benchmarks but not really planning benchmarks.



# Unaddressed Minor Concerns from Reviewers

- **Experiment limited to a Single Architechture**: The paper focuses on React style plan-act architecture. It is not immediately clear if this can be adopted to other architectures directly. (Reviewer v2FN)
- **Privacy/Security Concerns**: The cached plan may have privacy/security concerns, which are not addressed by the framework. (Reviewer RMgA )
- **Generality to other LLM families or smaller models**: The empirical evidence might not generalize to a different model family. (Reviewer v2FN)



# Recommendation Justification

The paper proposes a practically valuable and empirically validated technique for reducing serving costs of LLM‑based agents. Authors have addressed all major reviewer concerns with additional experiments and clarifications (Confirming with reviewer on this in a separate discussion). The remaining minor concerns (mentioned above) are natural avenues for future work and do not outweigh the contributions.

Given the strong reviewer support (average rating 4.2, one “accept” and four “borderline accept”s), the thorough rebuttal, and the relevance to the community, I believe the paper merits inclusion in the conference.